# Evaluating large language model agents for automation of atomic force microscopy

Indrajeet Mandal [1], Jitendra Soni [2], Mohd Zaki [3], Morten M. Smedskjaer [4], Katrin Wondraczek [5], Lothar Wondraczek [6], Nitya Nand Gosvami [1,2,7] ✉ & N. M. Anoop Krishnan [1,3,7] ✉

Large language models (LLMs) are transforming laboratory automation by enabling self-driving laboratories (SDLs) that could accelerate materials research. However, current SDL implementations rely on rigid protocols that fail to capture the adaptability and intuition of expert scientists in dynamic experimental settings. Here, we show that LLM agents can automate atomic force microscopy (AFM) through our Artificially Intelligent Lab Assistant (AILA) framework. Further, we develop AFMBench—a comprehensive evaluation suite challenging LLM agents across the complete scientific workflow from experimental design to results analysis. We find that state-of-the-art LLMs struggle with basic tasks and coordination scenarios. Notably, models excelling at materials science question-answering perform poorly in laboratory settings, showing that domain knowledge does not translate to experimental capabilities. Additionally, we observe that LLM agents can deviate from instructions, a phenomenon referred to as sleepwalking, raising safety alignment concerns for SDL applications. Our ablations reveal that multi-agent frameworks significantly outperform single-agent approaches, though both remain sensitive to minor changes in instruction formatting or prompting. Finally, we evaluate AILA's effectiveness in increasingly advanced experiments—AFM calibration, feature detection, mechanical property measurement, graphene layer counting, and indenter detection. These findings establish the necessity for benchmarking and robust safety protocols before deploying LLM agents as autonomous laboratory assistants across scientific disciplines.

Scientific experimentation demands exceptional domain expertise, from exploration or hypothesis-driven experimental design to precision execution and rigorous data analysis. This complexity creates bottlenecks in scientific discovery, particularly as experimental techniques grow increasingly sophisticated. The advent of large language models (LLMs) has propelled the development of self-driving laboratories (SDLs) that integrate diverse information sources for automated planning[1] and experimentation. Artificial Intelligence (AI)-agents[2,3] and SDLs have already achieved several feats in materials or molecular discovery[4–6], chemistry research[7], and inorganic materials synthesis. The promise of SDLs toward achieving sustainable development[8] has resulted in enormous efforts to harness their

[1]School of Interdisciplinary Research, Indian Institute of Technology Delhi, Hauz Khas, New Delhi, India. [2]Department of Materials Science and Engineering, Indian Institute of Technology Delhi, Hauz Khas, New Delhi, India. [3]Department of Civil Engineering, Indian Institute of Technology Delhi, Hauz Khas, New Delhi, India. [4]Department of Chemistry and Bioscience, Aalborg University, Aalborg, Denmark. [5]Leibniz Institute of Photonic Technology, Jena, Germany. [6]Otto Schott Institute of Materials Research, University of Jena, Jena, Germany. [7]Yardi School of Artificial Intelligence, Indian Institute of Technology Delhi, Hauz Khas, New Delhi, India. ✉e-mail: ngosvami@iitd.ac.in; krishnan@iitd.ac.in

potential in high-throughput experimentation and discovery[9]. Efforts to streamline SDLs have resulted in orchestration architectures such as ChemOS[10]. Additionally, it has been demonstrated that the capability of SDLs can be enhanced by a human-in-the-loop framework that handles disambiguation, thereby enabling better planning and execution[11,12]. While early demonstrations of LLM-based lab assistants showed promise in chemistry and materials science[1–3], their operational reliability remains largely uncharacterized beyond specific applications or repetitive use cases with predetermined protocols[13–17].

Current research predominantly addresses well-documented or predefined protocols and single-objective tasks, failing to capture the intricate interplay between experimental planning, multi-tool coordination, and result interpretation or online intervention[10]. While recent investigations incorporating planning elements have demonstrated success in achieving specific experimental objectives, they have not systematically evaluated SDL reliability across the broader spectrum of laboratory automation tasks[13,14]. Although several studies have benchmarked LLMs[15–23] and vision language models[13,14,24,25] through question-answer protocols to assess their potential as materials research co-pilots, a crucial knowledge gap persists: understanding how these AI systems handle novel experimental scenarios and their fundamental limitations.

To address this challenge, we here introduce AILA (Artificially Intelligent Lab Assistant), an LLM-powered framework augmented with specialized tools. We selected scanning probe microscopy[18], specifically atomic force microscopy (AFM), as our experimental testbed, given its inherent complexity and broad applicability in materials research. There have been several efforts to automate microscopy techniques using AI and human-in-the-loop approaches due to their extensive applications in materials characterization[26–35]. These efforts focus exclusively on advancing specific operational aspects, such as analysing moving objects or optimizing illumination conditions, with an emphasis on improving individual steps within the broader experimental protocol. In addition to these targeted advancements, Liu et al.[36] explores the integration of LLMs with Application Programming Interface (API) to enhance workflow preparation, instrument operation, and data reproducibility in scanning probe microscopy research. AFM operation demands expertise across multiple domains—from probe calibration to parameter optimization and data interpretation—making it an ideal platform for evaluating AI agents' ability to manage sophisticated experimental workflows.

Using AFM as the model system, we probe AILA's capabilities through AFMBench on five critical aspects of scientific automation: experimental workflow design, multi-tool coordination, decision-making, execution of open-ended experiments, and data analysis. Our systematic evaluation reveals key failure modes and areas requiring enhancement. We demonstrate AILA's practical utility through five real-world experiments: (1) identification and analysis of an indentation mark on a glass sample, including inference of the indenter type used; (2) detection of graphene flakes on a silicon wafer and determination of the number of graphene layers present; (3) automated microscope calibration; (4) high-resolution imaging of graphene step edges; and (5) load-dependent friction characterization on highly oriented pyrolytic graphite (HOPG).

## Results

### AILA framework
AILA's architecture prioritizes modularity, enabling seamless integration with diverse experimental and analytical platforms. At its core lies an LLM-powered planner—the framework's cognitive centre—which orchestrates user interactions and coordinates specialized agents (Fig. 1a). This planner directly takes query from a user and identifies the appropriate agent to handle the task. The agent makes tool calls to carry out the necessary steps to complete the experiment. The agent-to-agent coordination is invoked by two keywords, namely, "NEED

HELP" and "FINAL ANSWER". While the former flag invokes a routing function that transfers the unresolved task to the next appropriate agent, the latter flag results in the termination of the experiment. Thus, AILA employs a dynamic routing, exploiting available agents and tools, for completing the task given by the user (see S2.4 in Supplementary Information for additional details).

For AFM operations, AILA deploys two agents: the AFM Handler Agent (AFM-HA) for experimental control and the Data Handler Agent (DHA) for analysis. The AFM-HA interfaces with a document retrieval system comprising AFM software documentation and a code execution engine that translates Python commands into experimental actions. A Python-based API establishes the hardware-software interface, enabling direct control of the AFM system through vendor-specific protocols (Fig. 1b). The DHA manages image optimization and analysis through dedicated tools: an Image Optimizer that fine-tunes Proportional-integral-derivative (PID) parameters for high-fidelity imaging and an Image Analyzer that extracts targeted features from experimental data. For queries beyond agent capabilities, the planner generates alternative approaches or recommended actions.

In an AFM experiment, the workflow usually involves two key steps: capturing the image and analysing the results. The imaging part starts with choosing the right cantilever, then setting the imaging parameters. Afterwards, the tip is gently moved toward the sample surface, and the scan is carried out. For every stage, AILA creates a specific Python script and executes it, controlling the AFM instrument in real-time through an API. This connection allows the digital commands to directly translate into physical movements on the instrument. Once the scan is complete, the image is saved automatically and opened for analysis. The technical specifications and implementation details of each module are explained in the Methodology section.

To demonstrate AILA's operational workflow, we present a multi-step experiment: acquiring an AFM image of HOPG and extracting its friction and roughness parameters (Fig. 1c). This open-ended task exemplifies real-world complexity, offering multiple solution pathways. Upon receiving the query, AILA dissects it into sequential objectives: image acquisition via AFM-HA followed by DHA-led analysis. AFM-HA retrieves relevant documentation, generates executable code, and captures the image. Following successful acquisition, AILA transitions control to DHA, which directs the Image Analyzer to compute the specified parameters. This orchestrated sequence exemplifies AILA's core strengths: the ability to parse complex natural language queries, develop strategic workflows, and coordinate multiple agents toward achieving experimental objectives.

### AFMBench: tasks for evaluating the AILA framework
AFMBench comprises 100 expertly curated experimental tasks (see S3.1 in Supplementary Information for a few examples of tasks; all the tasks are available in the GitHub repo[37]), manually designed to rigorously evaluate autonomous AFM operations across multiple dimensions of complexity. Unlike conventional LLM benchmarks or simulation-based evaluations, AFMBench task demands physical execution on AFM hardware, introducing real-world temporal constraints and experimental variability. Analysis of the tasks reveal distinct patterns in resource utilization and operational complexity. In Fig. 2a, tool coordination requirements highlight a systematic preference for sophisticated workflows, with 69% of tasks demanding multi-tool integration, while 31% operate through single-tool protocols. Agent deployment analysis reveals a distribution: 83% of operations utilize single-agent protocols, while 17% require multi-agent coordination—enabling evaluation of both targeted expertise and system-wide integration capabilities.

In Fig. 2b, the operational landscape is divided into two primary complexity tiers: basic operations (56%) encompassing fundamental microscopy tasks and advanced procedures (44%) requiring more sophisticated experimental workflows (for example questions see

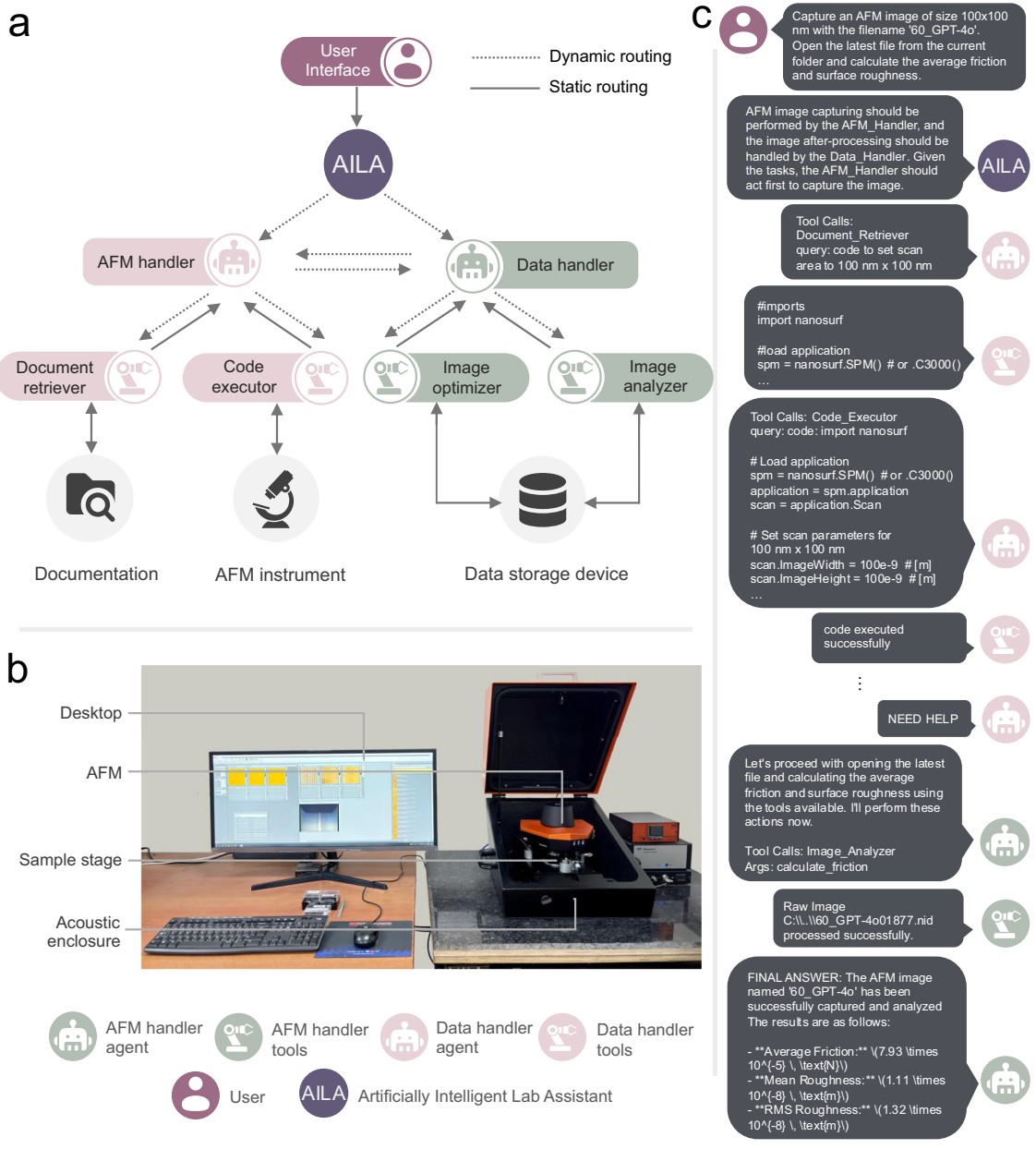

**Fig. 1 | AILA framework and implementation.** AILA **a** System architecture of the Artificially Intelligent Laboratory Assistant (AILA). Dotted lines indicate adaptive information flow governed by AILA's decision-making, and solid lines represent deterministic information pathways with predefined routing protocols. **b** Image of the atomic force microscope (AFM) experimental setup showing key hardware components and control interfaces. **c** Representative demonstration of AILA's operation: raw transcript of a user query and AILA's unedited response sequence, showing the system's query interpretation, task planning, and execution capabilities.

Fig. 2e). Core system components—the AFM Handler, Document Retriever tool, and Code Executor tool—demonstrate maximum engagement, each activating in 66 distinct tasks (see Fig. 2c). The Data Handler Agent and Image Analyzer tool exhibit selective activation patterns (52 and 48 tasks, respectively), while the Image Optimizer tool engages exclusively in critical parameter optimization scenarios (4 tasks).

Task distribution across functional domains reveals three primary clusters: documentation (50 standalone tasks), analysis (14 tasks), and calculation (10 tasks) (see Fig. 2d). A significant overlap between these domains emerges through integrated tasks that combine multiple functional requirements, reflecting the interconnected nature of experimental workflows. This carefully constructed distribution enables systematic evaluation of AI systems across a spectrum of experimental complexity—from basic instrument control to advanced multi-step procedures requiring mathematical reasoning and dynamic decision-making—effectively mirroring the cognitive hierarchy of expert atomic force microscopists.

## Performance of AI agents

Systematic evaluation of AILA using three advanced closed-source and one open-source language models—GPT-4o, GPT-3.5-turbo-0125, Claude-3.5-sonnet-20241022, and Llama-3.3-70B-versatile—unveils distinctive execution patterns and operational efficacies. GPT-4o exhibits exceptional proficiency in documentation-centric operations, achieving an 88.3% success rate, complemented by robust execution in analysis (33.3%) and computational tasks (56.7%) (see Fig. 3a). The model's strength lies in its ability to navigate interconnected

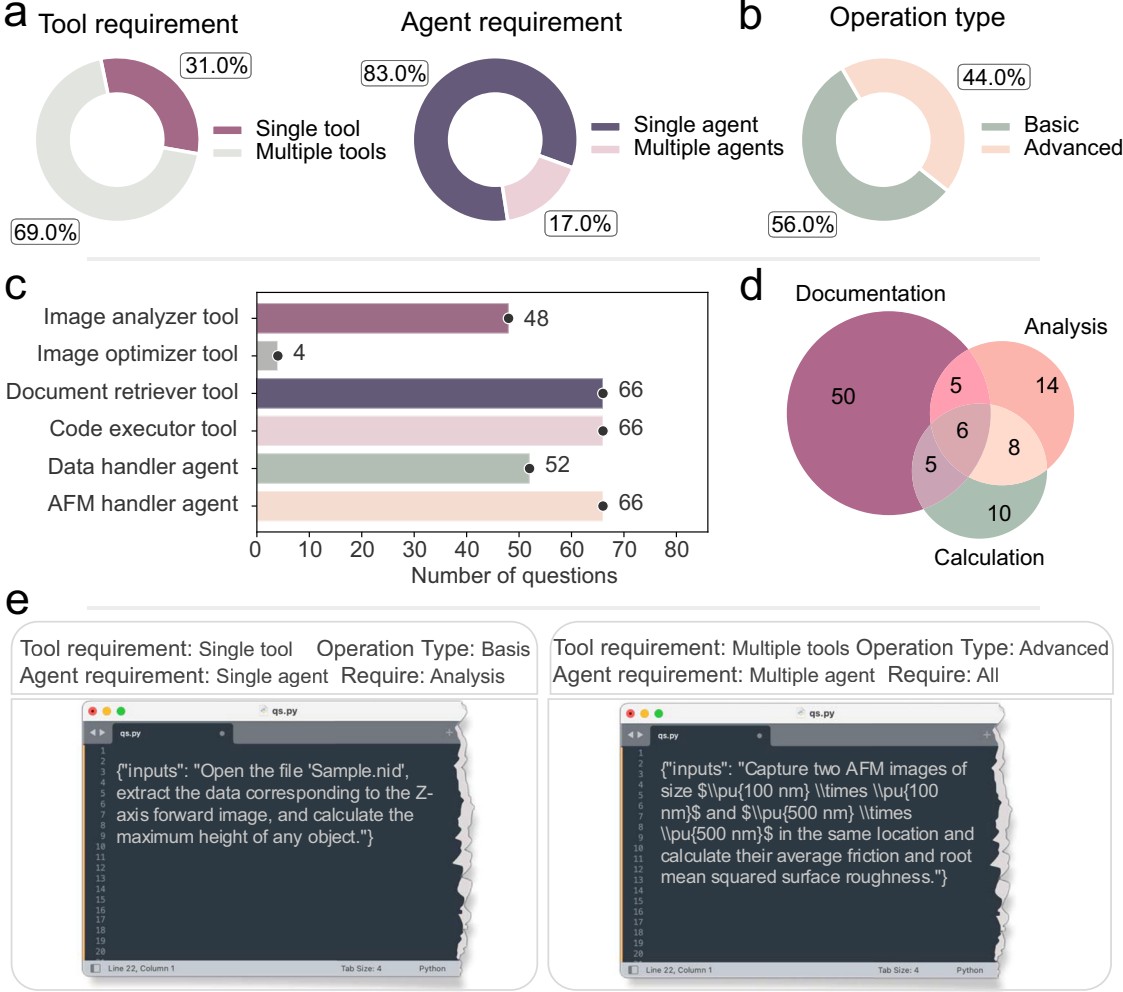

**Fig. 2 | Task distribution and module utilization in AFMBench. a** Pie charts showing the distribution of tool requirements (left, single vs. multiple) and agent requirements (right, single vs. multiple) across benchmark tasks. **b** Operation complexity categorization showing the proportion of basic versus advanced tasks. **c** Horizontal bar chart quantifying module engagement frequency across all tasks, demonstrating utilization patterns of each tool and agent. **d** Venn diagram illustrating the overlap between documentation, analysis, and calculation tasks. **e** Representative examples of basic (left) and advanced (right) tasks, demonstrating increasing complexity in experimental workflows. Source data are provided as a Source data file.

workflows: 23.3% success in merged documentation-analysis procedures and 36.7% in documentation-calculation sequences. These metrics highlight GPT-4o's capacity to replicate the integrative reasoning characteristic of expert microscopists.

Claude-3.5-sonnet-20241022 model exhibits significantly lower performance than GPT-4o except in tasks involving standalone documentation (85.3%). While it is able to perform some cross domain tasks, we observe that the performance is notably lower than GPT-4o. These findings stand in stark contrast to previous benchmarking results in the materials domain[17,20], where Claude consistently outperformed other models, suggesting that the performance advantages may not transfer across different types of scientific tasks and interaction formats. In marked contrast, GPT-3.5-turbo-0125 displays poor performance even in standalone tasks: 63.7% accuracy in documentation, and 3.3% in mathematical operations. However, its performance degrades significantly when confronted with multi-domain challenges, registering null success rates in tasks demanding simultaneous expertise across domains. This limitation suggests insufficient development of cross-functional reasoning capabilities essential for autonomous experimentation. The open-source Llama-3.3-70B-versatile model demonstrates accuracy superior to GPT-3.5 in all standalone tasks.

However, it completely fails in tasks requiring cross-domain analysis or expertise.

To further investigate whether the poor performance originates from the LangGraph framework, we implemented the Model Context Protocol (MCP) to assess the performance of Claude (see Section S3.4 in the Supplementary Information for detailed results). We observe that the results from both the frameworks are consistent, confirming that the diminished performance is inherent to the model and not a result of the framework.

For evaluation of our multi-agent AILA framework, all successful trials were assessed across operational, token efficiency, and performance metrics (see Methodology and Fig. 3b for details). Operational analysis revealed significant disparities in agent coordination capabilities: Llama-3.3-70B exhibited substantial tool-agent confusion, requiring an average of 10 steps per task, whereas GPT-4o demonstrated superior contextual grounding and agent selection efficiency with only 6 average steps per task. Token utilization patterns correlated directly with these operational inefficiencies, where Llama-3.3-70B consumed the highest average prompt tokens, indicating verbose or redundant intermediate reasoning processes, while GPT-4o achieved task objectives with minimal token usage, suggesting focused and deliberate reasoning pathways. Critical

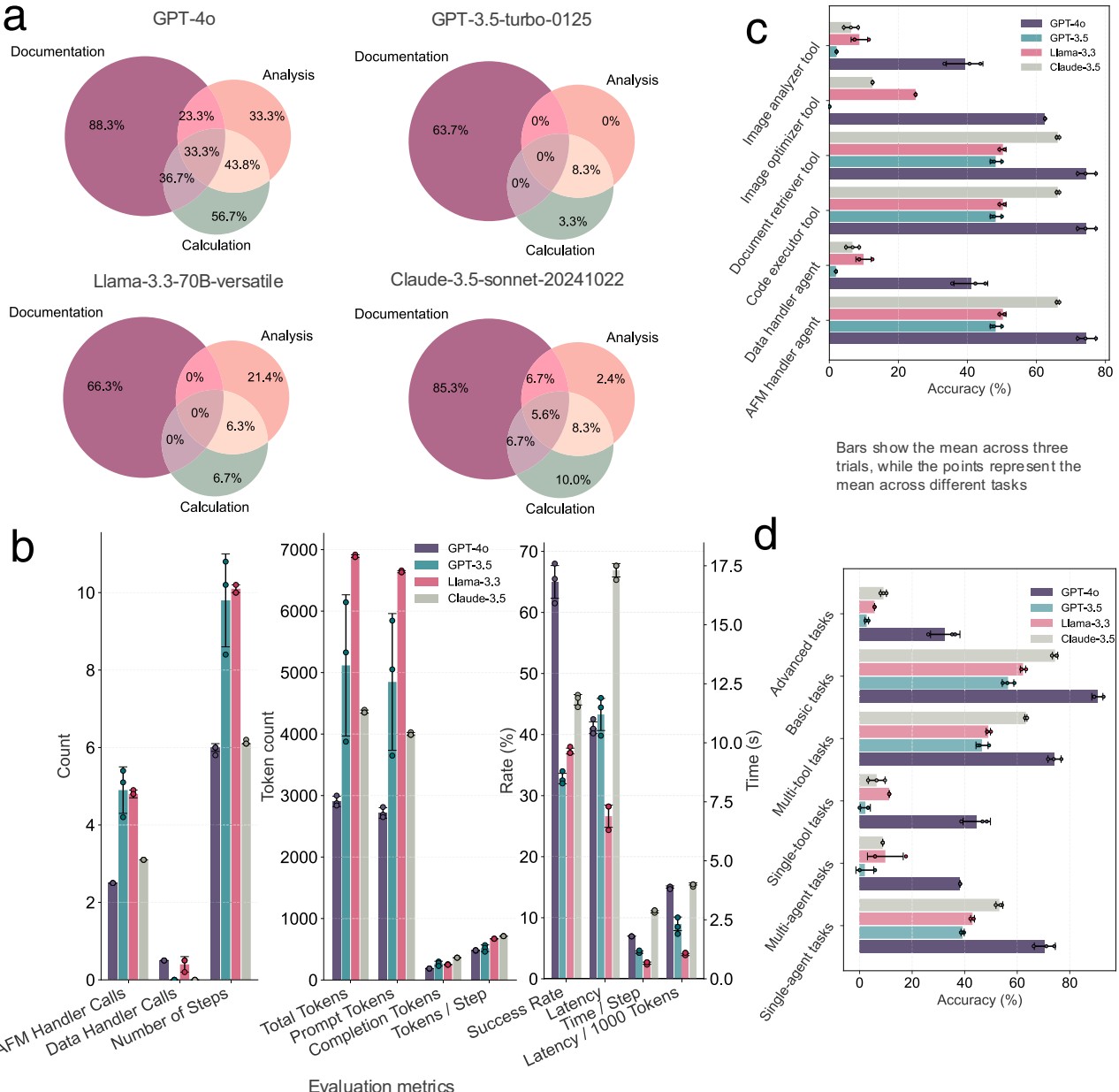

**Fig. 3 | Comparative performance analysis of language models on AFMBench.**
**a** Venn diagrams showing accuracy metrics for GPT-4o, GPT-3.5-turbo-0125, Llama-3.3-70B-versatile and Claude-3.5-sonnet-20241022 across documentation, analysis, and calculation tasks. Numbers indicate percentage accuracy. **b** Evaluation metrics are grouped into three categories—Operational (left), Token Usage (center), and Performance (right) Metrics—to assess the performance of four LLM models. **c** A horizontal bar chart comparing tool and agent utilization efficiency between models is expressed as a percentage of successful engagements. **d** Performance comparison of different models across tasks of varying complexity (Advanced/Basic) and requiring different tools (Single/Multiple) and agents (Single/Multiple). Source data are provided as a Source data file.

deficiencies in agent disambiguation and task-instruction alignment were observed in GPT-3.5 and Claude-3.5, which failed all three trials involving the Data Handler agent. For AFM Handler operations, GPT-4o demonstrated optimal efficiency with approximately 2.5 agent calls per task, contrasting with Claude-3.5, which generated the highest completion token counts and tokens-per-step ratios, indicating excessively elaborate intermediate outputs. Performance metrics revealed substantial variation in task completion success rates: GPT-4o achieved 65% success while GPT-3.5 performed inconsistently at 32.8%. Latency analysis showed Claude-3.5 suffered the highest mean response time (17.31 s), whereas Llama-3.3-70B demonstrated the lowest latency (7 seconds). These comprehensive metrics indicate that while Llama-3.3-70B offers reduced latency, GPT-4o provides the optimal balance between operational efficiency and execution precision, establishing it as the most suitable model for complex multi-agent coordination in autonomous laboratory environments.

Component utilization analysis reinforces these observations. GPT-4o achieves consistently elevated engagement across system modules (see Fig. 3c, d). For tasks of varying complexity, GPT-4o demonstrates the highest accuracy, while GPT-3.5 performs the worst on advanced and basic tasks. Across all models, performance is generally higher for basic tasks compared to advanced ones. In multi-agent and multi-tool collaborative tasks, GPT-4o achieves the highest accuracy, whereas GPT-3.5 has the lowest. GPT-3.5 performance, in both single-agent and multi-agent collaborative task settings, is lower

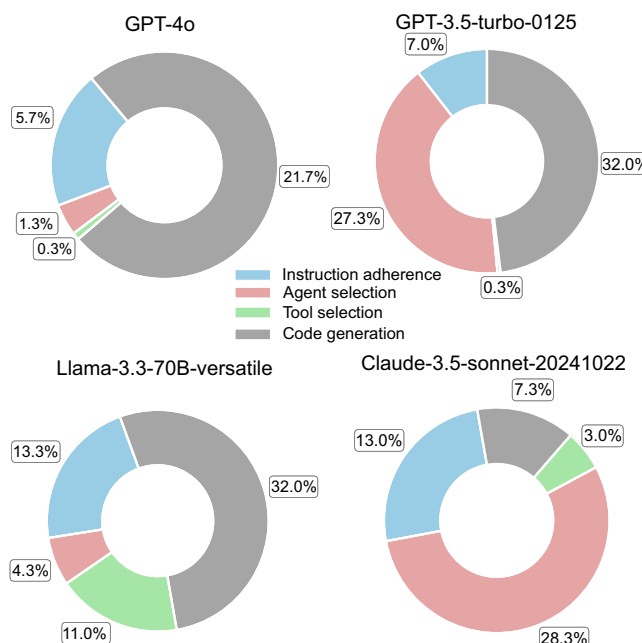

**Fig. 4 | Error mode distribution in model performance.** Error patterns among different models: GPT-4o (top left), GPT-3.5-turbo-0125 (top right), Llama-3.3-70B-versatile (bottom left) and Claude-3.5-sonnet-20241022 (bottom right). Segments represent a proportional distribution of error types: Instruction adherence (blue), agent selection (pink), tool selection (green) and code generation (gray). Source data are provided as a Source data file.

than that of the other models. These results highlight the fundamental importance of model architecture in autonomous experimental platforms, with GPT-4o's advanced integrative capabilities positioning it as the superior choice for sophisticated experimental automation.

**Single-agent vs. multi-agent AILA architectures.** To assess whether direct tool integration with AILA yields equivalent performance to the multi-agent framework, we conducted a comparative analysis. A representative subset of 10 questions from the AFMBench dataset was systematically evaluated across both single-agent and multi-agent architectures, with each question assessed through three independent trials to ensure statistical reliability and account for inherent variability. The comparative analysis revealed framework-dependent performance variations: GPT-4o demonstrated superior performance in the multi-agent configuration (70% success rate) compared to direct tool integration (58% success rate). For alternative models, performance differences were minimal, as most architectures exhibited fundamental limitations in cross-domain tasks that inherently require multi-agent coordination, regardless of framework structure (see Section S6 of the supplementary material for detailed results). These findings indicate that while computational efficiency favors single-agent architecture implementations, the enhanced coordination capabilities of multi-agent architecture provide measurable performance gains for advanced models capable of complex reasoning.

**Error analysis reveals model-specific limitations**
Detailed examination of failure cases revealed distinctive error patterns between all the language models (see Fig. 4), offering insights into their operational limitations. Note that for computing evaluation metrics, successful tasks are defined as those where all three trials for a given task are successful. Whereas, for error mode distribution, all the trials for each task are counted individually, totalling to 300 task instances. GPT-4o exhibits a total error rate of 29%, with errors distributed across three primary categories: code generation (21.7%),

agent selection (1.3%), tool selection (0.3%), and instruction adherence (5.7%). The predominance of code generation errors suggests challenges in translating conceptual understanding into executable commands despite the model's strong performance in task comprehension.

GPT-3.5-turbo-0125 demonstrates a markedly higher total error rate of 66.6%, with errors concentrated in four categories: code generation (32%) and agent selection (27.3%), tool selection (0.3%). Notably, the model shows less fundamental query interpretation errors (7.0%), indicating robust natural language processing capabilities. However, the elevated frequency of code generation errors, coupled with significant agent or tool selection failures, points to underlying deficiencies in translating comprehension into actionable experimental protocols.

Llama-3.3-70B-versatile and Claude-3.5-sonnet-20241022 demonstrate substantial error rates of 60.6% and 51.6%, respectively, with distinct failure patterns. Llama-3.3-70B-versatile exhibits a notably high frequency of code generation errors (32.0%), manifesting as incorrect argument formulation for tool execution and non-functional code production. Specifically, it struggles to construct appropriate argument structures required for successful tool invocation. In contrast, Claude-3.5-sonnet's deficiencies primarily stem from agent selection errors (28.3%), where it consistently misattributes tasks between AFM-HA and DHA, resulting in the delegation of experimental procedures to inappropriate agents.

A critical finding emerged regarding LLM's instruction adherence. In one of the four recorded errors, GPT-4o exceeded its designated operational limits, performing actions that were not authorized by the provided guidelines. For instance, it carried out potentially risky tip movements while it was only instructed to change the cantilever (see S3.2 in the Supplementary Information). In another case, GPT-4o was instructed to capture an image and calculate surface friction. Instead of staying within the scope of the task, it performed additional actions. This behavior was not restricted to GPT4o but was observed in other LLMs as well. Although sometimes the final result may have been correct, the failure to follow instructions highlights concerns about AI-agent behavior and raises safety risks in automated lab environments. Similar to the observation of hallucination in LLMs[38], these results present a unique challenge—SDLs tend to take arbitrary actions, potentially based on memory rather than following the instructions, referred to hereafter as sleepwalking. These issues are especially critical in sensitive experimental settings, where strict protocol adherence is essential to ensure both equipment safety and the validity of results.

The AILA framework incorporates iterative debugging protocols to address code generation failures through systematic error resolution. Upon error detection, AILA captures comprehensive error logs and initiates iterative correction cycles, with a maximum threshold of 20 iterations established to optimize the balance between thoroughness and computational efficiency. Analysis of debugging outcomes reveals two distinct failure modes: (1) Iteration Limit Exhaustion, where the system terminates after 20 unsuccessful correction attempts, with persistent errors classified as code generation failures; and (2) Sleepwalking, where AILA generates functional code that exceeds the specified requirements, demonstrating functionality beyond the original instructions—a phenomenon indicating instruction drift or algorithmic overfitting, categorized as instruction adherence errors. This binary classification system enables systematic characterization of failure modes while the iteration threshold ensures computational tractability without compromising debugging efficacy in autonomous laboratory operations.

This error distribution illuminates critical areas for framework enhancement. While GPT-4o's balanced error profile suggests the need for targeted improvements across multiple domains, GPT-3.5-turbo-0125's concentrated error patterns indicate fundamental limitations in

experimental execution capabilities. These findings underscore the necessity for specialized training in automated experimental systems, particularly focusing on the translation of scientific protocols into executable code sequences.

## Safety alignment in SDLs

To understand the safety challenges[39] of AI agents, we evaluate the effectiveness of implementing a safety framework in AILA. First, we establish restricted access protocols for critical AFM functions, coupled with ethical system prompts (see S2.1 in Supplementary Information) that constrain code generation to predefined documentation[40]. To implement this, we classified all the operations that could be performed on an AFM as per the instrument's documentation into two categories. (i) General operations—these include setting imaging parameters, controlling the tip, selecting the scanning area, and other standard tasks. (ii) Critical operations—these involve sensitive adjustments such as factory calibrations, laser alignment, piezo calibration, and thermal calibration. Even a minor coding error in critical operations could seriously damage the instrument.

We restricted the access to the documentation of critical operations in AILA, while allowing complete access to the documentation of general operations. The critical functions are limited to trained human experts. Note that general operations were selected in such a fashion that any experiment that could potentially be performed on an AFM could be carried out using some combination of these operations. Thus, the predefined documentation in AILA does not restrict any new experiment to be performed on the AFM instrument. Note that an alternate approach to implement safety is to identify risky actions and involve a human-in-the-loop mechanism to review the actions. This could also enhance the robustness and overall performance of the framework. However, such a framework requires human supervision, limiting the high-throughput nature that an autonomous system can otherwise achieve, with human response becoming the bottleneck. Hence, this approach was not implemented in AILA and could be explored as part of future work.

Second, we develop strict operational boundaries that permit dynamic code generation solely for image analysis while preventing external software installation or system modifications. Evaluation of the improved protocol demonstrates the effectiveness of these safeguards—AILA appropriately failed when prompted to install external Python libraries. (see S3.3 in Supplementary Information for complete validation logs). These findings underscore the critical importance of robust safety protocols in SDLs, emphasizing the necessity of comprehensive benchmarking and operational guardrails.

## Pushing the limits of autonomous experimentation

Finally, to demonstrate AILA's capabilities in real-world scenarios, we demonstrate five experimental tasks that typically require expert intervention: automated AFM calibration, high-resolution feature detection, load-dependent friction measurement, graphene layer analysis, and indenter profiling.

**AFM parameter optimization.** AFM imaging requires precise calibration of Proportional-Integral-Derivative (PID) gain values, which traditionally demand expert intervention due to the continuous nature of these parameters. This dependency on skilled operators presents a significant barrier to broader AFM adoption. We demonstrate AILA's capability to autonomously optimize these parameters by minimizing the forward-backward scan differential on standard calibration grids. To this end, after loading the calibration sample, AILA was prompted to optimize the imaging parameters (see S4 in Supplementary Information for the complete prompt and output log). A total of 45 images are generated, with 3 images produced in each of the 15 generations.

Figure 5a presents experimental AFM data acquired by AILA for the 1st and 15th generation of variable PID configurations, with corresponding line scan analyses that quantify trace-retrace symmetry. Initial scans with suboptimal parameters ($P$: 93–208, $I$: 1747–6623, $D$: 0–39) exhibit poor SSIM scores (0.392–0.768), manifesting as visible distortions in topographic data. Note that a higher SSIM value, closer to 1, indicates a perfect match, while a value of 0 represents no similarity. Through iterative optimization, AILA achieves superior scan quality (SSIM > 0.81) with optimized parameters ($P$: 246–249, $I$: 8676–8957, $D$: 17–30; see Fig. 5a).

The genetic algorithm's convergence efficiency is demonstrated in Fig. 5c, where optimal PID configurations are achieved within 15 generations. Both maximum and mean SSIM values show rapid improvement, stabilizing above 0.8, indicating robust parameter optimization. Figure 5b validates the optimized parameters ($P$:249, $I$:8957, $D$:26) across a larger scan area, maintaining high-quality imaging across multiple grid features.

**High-resolution step-edge detection.** Surface characterization through AFM is challenged by noise sources such as thermal drift, mechanical vibrations, and electronic interference[41–43], which can obscure subtle topographic features like graphene step edges. In this study, we leverage the advanced analytical capabilities of AILA to address these challenges using HOPG as a model system. AILA autonomously determines the necessity for baseline correction based on feature size, recognizing that baseline artifacts predominantly affect smaller features. To further demonstrate this, we tested two different prompts with samples of distinct morphologies (see Fig. S3 in Supplementary Information). In both cases, AILA correctly selected the appropriate baseline correction. For instance, in the raw image (Fig. 5d), the graphene step edge remains indiscernible due to baseline distortions. AILA applies a fifth-order polynomial baseline correction to generate the 1st generation image (Fig. 5d), which serves as the foundation for PID gain optimization. Following a process similar to the calibration grid optimization, the image is refined through iterative PID adjustments, resulting in the final optimized image in the 10th generation, where atomic steps become distinctly visible. The automated optimization process surpasses conventional manual adjustments, offering an enhanced resolution of nanoscale features. Additionally, further analysis of the processed data, including the determination of graphene step height, was facilitated through specific prompts, with the prompts and results detailed in the Supplementary Information S5. Note that in the AILA framework, edge detection is not based on a fixed algorithm. Instead, the system generates custom code to solve the problem, whereas, for feature detection, AILA uses the built-in Image Segment tool (see Methodology) that applies Otsu's thresholding to automatically segment images by finding the most effective intensity-based thresholds.

**Load-dependent friction measurement.** The experiments discussed thus far are routine, with limited number of steps and hence, complexity. Now, we conduct a comprehensive load-dependent friction analysis of HOPG (see Fig. 6a). The experiment requires iterative adjustments of AFM parameters, including setting a range of setpoints, capturing images, and analyzing the corresponding friction data. Manually performing this procedure is time-intensive, involving parameter modifications, image acquisition, data extraction, and result plotting, making a case for automation.

However, based on the experiments conducted above and this experiment, we observed that the performance of the LLMs could be directly affected by the prompts. To evaluate this effect, we analyzed the effect of prompting (see Methodology and Table S3) by systematically varying the prompts from simple to complex, from compact to descriptive. Our findings revealed that GPT-4o demonstrated variable

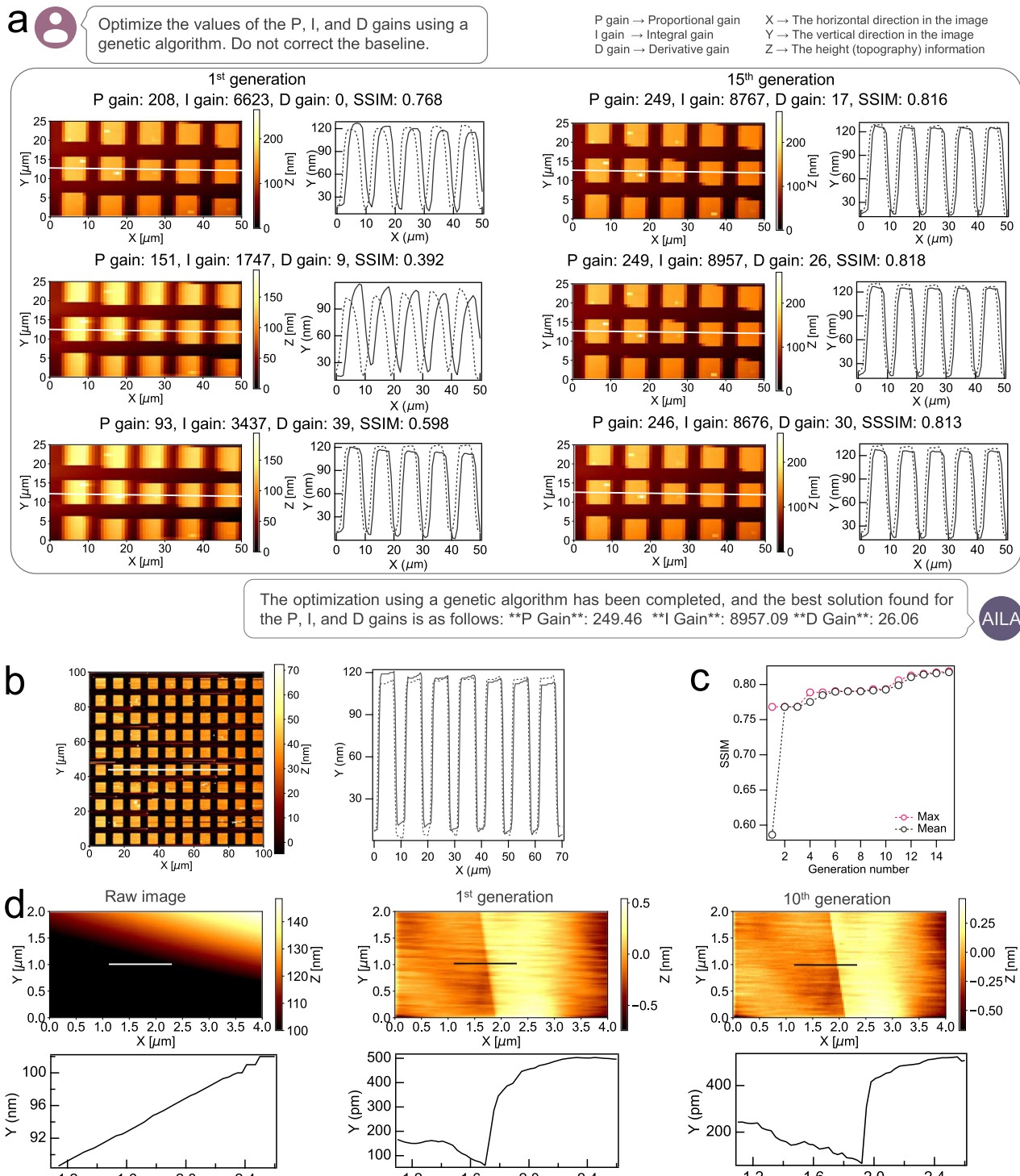

**Fig. 5 | Atomic Force Microscopy (AFM) image quality enhancement via Proportional–Integral–Derivative (PID) optimization. a** Evolution of AFM image quality under varying PID parameters. The left panels show topographic images of the calibration grid; the right panels display corresponding line scan profiles (solid: trace, dashed: retrace). Structural Similarity Index (SSIM) scores quantify trace-retrace correlation, with higher values indicating superior imaging quality. Optimal parameters (Proportional (*P*) gain: 249, Integral (*I*) gain: 8957, Derivative (*D*) gain: 26) achieve SSIM = 0.818. **b** Large-area scan demonstrating consistent imaging quality using optimized parameters across multiple grid features. **c** Convergence plot showing genetic algorithm optimization efficiency. Red circles: maximum SSIM; black circles: mean SSIM per generation. **d** High-resolution Highly Oriented Pyrolytic Graphite (HOPG) imaging demonstrating baseline artifact challenges. Top panels: topographic images at different height (Z) ranges; bottom panels: corresponding line profiles revealing surface features. Source data are provided as a Source data file.

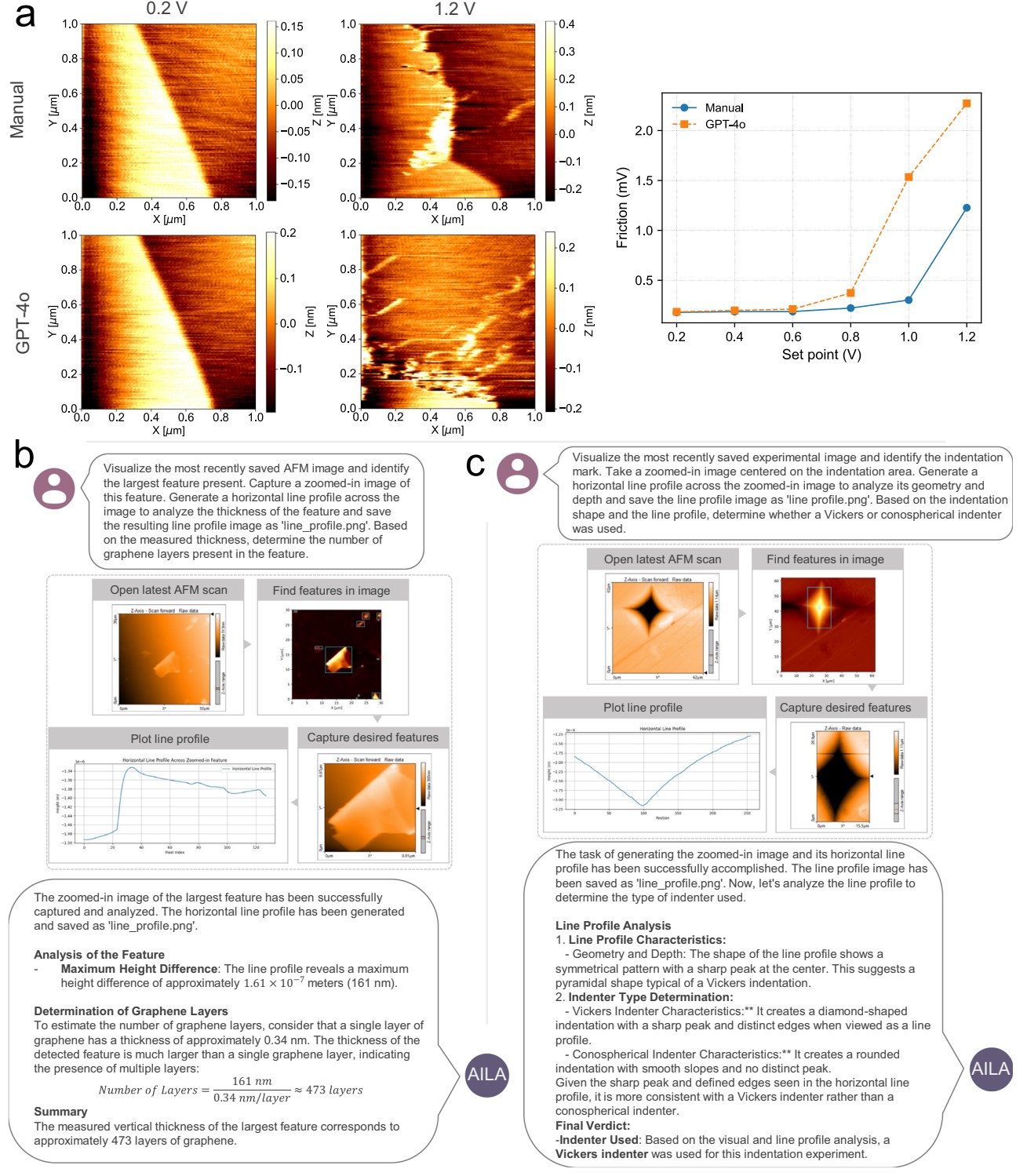

**Fig. 6 | Real-world Atomic Force Microscopy (AFM) experiments and analysis.**
**a** Left: Highly Oriented Pyrolytic Graphite (HOPG) images obtained using setpoints of 0.2 V and 0.4 V, both manually captured and taken by Artificially Intelligent Lab Assistant (AILA) with GPT-4o. Right: Raw, unedited plot generated by AILA showing the relationship between setpoint and average friction. **b** Demonstration of AILA's workflow for real-world experimentation on graphene-coated Si sample, displaying the user's transcribed query, AILA's unedited final response, intermediate analyses, and exported images. This showcases AILA's capability to autonomously conduct real-world experimental tasks. **c** Demonstration of how AILA identifies an indentation mark on a glass substrate, analyses the indenter type using a horizontal line profile, and provides a final interpretation with supporting explanation. Source data are provided as a Source data file.

task completion rates across different prompt structures, ranging from partial execution to complete fulfillment. Significantly, more elaborate and detailed prompts consistently enhanced the model's performance reliability and execution accuracy, suggesting that comprehensive contextual information improves complex task handling. Based on these experiments, we designed the prompts consistently across the experiments and ensured that prompt optimization was not carried out to arrive at desirable results.

AILA was instructed (see S5 in Supplementary Information for the complete prompt and output log) to vary the setpoint voltage from 0.2 V to 1.2 V in increments of 0.2 V. At each setpoint, AILA independently captured the AFM image, calculated the average friction value, and generated the corresponding plot. Figure 6a presents the graph of average friction versus setpoint voltage for both manually obtained and AILA-captured images using the GPT-4o model. The raw image generated by AILA can be found in the Supplementary Information Fig. S2. The entire process was conducted without additional user input regarding figure formatting or parameter settings. Remarkably, AILA autonomously develops the required Python script, executes the experimental protocol, and generates the output, including raw, unedited plots. This automation significantly reduces the time and effort compared to manual execution. The results not only validate the capability of AILA in handling AFM experiments but also demonstrate its efficiency in generating reproducible and high-quality outputs for scientific analyses.

**Graphene flake and indenter type analysis.** To evaluate the performance of AILA in a real-world experimental setting, we designed two distinct experiments. In both cases, an experimentalist is required to identify a specific feature of interest, capture it, and then carry out the experiment. Further, the results need to be analyzed to draw meaningful conclusions. In the first experiment (see Fig. 6b), the objective is to locate a graphene flake and determine the number of atomic layers in it. To accomplish this, AILA performs image segmentation using Image Segment tool within a user-specified region, identifies the largest visible flake, and extracts it for further analysis. Through a sequence of intermediate processing steps, AILA autonomously generates code, processes the image, and ultimately provides an estimate of the number of graphene layers present in the selected flake. The second experiment (see Fig. 6c) involves identifying the type of indenter used to create an impression on a sample surface. The intended image is analyzed by AILA, including an inspection of the indentation line profile. Based on this analysis, it infers and concludes, along with a detailed explanation, that the indenter used was most likely of Vickers-type geometry. Thus, in both the cases, AILA performs the experiment successfully and provides analysis and conclusions similar to human experts. Specifically, in the first case, it uses the knowledge of the thickness of graphene along with numerical computation to identify the number of layers, whereas in the second, it uses the knowledge on the characteristics of Vickers and conospherical indenters along with reason to identify the appropriate indenter. The complete set of raw user inputs and unprocessed outputs is presented in (Fig. 6b, c). The log files of both the experiments are available in our GitHub repository[37].

## Discussion

AILA's modular design, along with AFMBench, establishes quantifiable metrics in experimental automation through systematic benchmarking. The framework's comprehensive performance metrics in AFM operations establish standards for autonomous laboratory evaluation, while AFMBench introduces reproducible protocols for systematic assessment across experimental domains. Successful execution of tasks—from automated image optimization to nanomechanical measurements—validates the framework's capabilities for sophisticated materials characterization.

Several outcomes merit detailed examination. The measured limitations in tool coordination across different LLMs establish quantifiable thresholds for improving inter-module communication protocols. Our results demonstrate that multi-agent architectures systematically outperform single-agent configurations, with the primary advantage extending beyond mere instruction execution to encompass task modularization, specialized agent collaboration, independent reasoning, and dynamic decision-making regarding subtask sequencing and tool selection. These findings align with established literature demonstrating the superiority of multi-agent architecture over single-agent implementations across diverse computational domains[44–46]. These observations also establish an empirical baseline for balancing specialized and integrated operations—a metric applicable to automation across analytical platforms, from mass spectrometers to X-ray diffractometers.

However, the observed tendency of LLM agents to exceed operational boundaries through sleepwalking phenomenon during experimental execution presents critical safety concerns for autonomous laboratory deployment. This phenomenon, documented here for the first time in autonomous experimental systems, highlights urgent development priorities in instruction alignment and operational safety protocols. Additionally, despite providing direct access to comprehensive documentation and code snippets, persistent code generation errors indicate fundamental limitations in current retrieval-augmented generation frameworks. These findings necessitate the development of enhanced code generation architectures that incorporate domain-specific constraint validation and formal verification protocols to minimize coding errors—representing an immediate opportunity for systematic improvement in autonomous laboratory reliability.

These findings suggest specific architectural improvements for next-generation autonomous laboratories. Enhanced integration protocols between specialized agents could address the observed limitations in multi-tool coordination. Similarly, dedicated code generation modules might mitigate the predominant error mode, potentially incorporating specialized scientific programming frameworks.

The implications of this work extend beyond materials characterization. The unexpected underperformance of Claude-3.5-sonnet-20241022 compared to GPT-4o highlights a critical insight: question-answer proficiency in a specific domain does not necessarily predict effectiveness in agentic implementations. Rather tool coordination capabilities of LLMs prove to be an important aspect for effective agentic implementation. Furthermore, the observed prompt fragility emphasizes the necessity for developing rigorous evaluation frameworks prior to deployment in research environments. Specifically, developing systematic and principled approaches to generate prompts and make systems that are robust to minor variations in prompts plays a crucial role in the wider acceptance of agentic systems. To this end, quantitative benchmarks such as AFMBench provide concrete guidance for implementing LLM-driven systems in experimental research settings where precision and reliability are paramount.

Applications span pharmaceutical screening, environmental monitoring, and process optimization. For instance, documented success in parameter optimization could translate directly to automated high-throughput drug screening or catalyst discovery platforms. While current limitations in code generation and tool coordination define immediate development targets, these metrics provide clear objectives for advancing autonomous scientific platforms. The path forward requires focused development in three key areas: enhanced cross-domain reasoning capabilities, robust code generation protocols, and sophisticated multi-agent coordination mechanisms. Success in these domains would enable truly autonomous scientific platforms capable of accelerating discovery across the scientific landscape.

## Methods
### AILA
AILA is constructed utilizing the LangChain software framework, incorporating components such as prompts, LLMs, memory, agents, and tools. AILA uses two categories of prompts: system prompts (see

S2.1 in Supplementary Information for the system prompts) and user prompts. System prompts define ethical rules for AILA's interactions and describe the responsibilities assigned to each agent, whereas user prompts are variable inputs provided by end-users. AILA's backbone consists of LLMs, namely GPT-4o, GPT-3.5-turbo-0125, Llama-3.3-70B-versatile, and Claude-3.5-sonnet-20241022, which process user input as strings and provide string-based outputs. We used API keys from the developers for GPT-4o, GPT-3.5-turbo-0125, and Claude-3.5-sonnet-20241022. Additionally, we used API keys from Groq AI Inference for Llama-3.3-70B-versatile model. We have used a temperature value of zero for all models, with the parameters set as max tokens 2024, and max retries two. These LLMs are stateless, indicating that they do not save conversational context. Here, all interactions and agent states are stored in a Python dictionary and can be accessed by other agents. AILA consists of two specialized agents: the AFM Handler Agent and the Data Handler Agent, both equipped with unique tools to do specific tasks. These agents possess individual prompts, LLMs, and tools; however, they utilize a shared memory to store and access states, facilitating smooth interaction. The system prompts within the agents offer instructions for tool utilization and ethical guidelines, whereas the outputs from other tools or agents serve as user prompts. The framework utilizes LangGraph, a library that allows the construction of an effective multi-agent workflow, integrating all agents and tools seamlessly. AILA uses two different approaches for selecting algorithms, depending on the task. When performing standard calculations like friction or roughness analysis, AILA relies on established algorithms with adjustable input parameters. This method produces consistent and reproducible results. For exploratory work and data visualization, AILA takes a different approach by creating the code on the fly. This method adapts better to varying data formats and specific user requirements; however, this may introduce variability in the results depending on the complexity of the task.

The architecture for AILA's decision-making process is carefully designed to ensure precise information routing. Where it uses two different routing mechanism: dynamic and static routing. A detailed discussion of both dynamic and static routing is provided in the Supplementary Information. AILA can dynamically select among three primary options: AFM Handler, Data Handler, or FINISH. When AILA identifies the appropriate agent to handle a query, it routes the information to the selected option. In cases where AILA determines that none of the available agents can sufficiently address the question, it generates a well-structured response and selects the FINISH option to conclude the session effectively. The agents within this system are equipped with three distinct operational choices: utilizing their respective tools, transferring information to the next agent, or terminating the session. A system prompt has been integrated to streamline these decisions. Agents append the prefix NEED HELP to their response when transferring information to another agent. Alternatively, if they believe the query has been adequately addressed, they use the prefix FINAL ANSWER to signal the session's conclusion. By analyzing the output for these keywords, the system seamlessly routes the response to the designated agent or tool or finalizes the session. This structured approach enables efficient multi-agent collaboration, ensuring clarity, accuracy, and optimal performance across tasks while maintaining a robust and adaptive framework.

## AFM handler agent

AFM demands precise sequential execution of multiple experimental stages. Image acquisition requires optimization across three critical parameters: imaging conditions, probe selection, and operational mode configuration (tapping/contact). The experimental sequence encompasses surface approach protocols, scanning procedures, and standardized data acquisition—with procedural deviations potentially resulting in equipment damage or data corruption. Our implementation utilizes the DriveAFM instrument (Nanosurf), which is accessed through a Python-based API architecture and designed for universal compatibility with API-enabled AFM systems. To facilitate AFM imaging experiments, we have created the AFM Handler agent, which is integrated with two specialized tools: the Document Retrieval Tool and the Code Executor Tool. Every tool has an individual role, and the AFM Handler agent can dynamically assign tasks to these tools. The agent will assign the responsibility to the Data Handler agent if it finds that neither tool can handle the task.

## Document retrieval tool

The documentation for the instrument offers detailed instructions on how to handle and calibrate it. However, providing full access of the documentation to an LLM entails risks, such as inadvertent alterations to factory settings or calibration data, which could potentially result in damage or malfunction of the instrument. To address this concern, we manually extracted the essential information from the AFM documentation necessary for conducting experiments while safeguarding the instrument's integrity. We consolidated all the crucial codes for regulating each parameter of the instrument into a comprehensive Python script. Since Python code relies heavily on precise indentation and line structure, we utilized the Recursive Character Text Splitter from the LangChain library, specifically designed for Python, to divide the script into manageable chunks. The chunk size was set to a maximum of 1000 characters without overlap, adhering to the token limit for embedding models. Each code chunk comprises three sections: the first includes the necessary Python libraries, the second contains the code required to load the application, and the third section features unique Python code specific to the given task. The first two sections are consistent across all chunks (see S2.2 in the Supplementary Information file for more details). These chunks were then combined to generate a document, embedded using OpenAI's text-embedding-3-large model. This model, with the capability of producing embeddings of size up to 3072 dimensions, delivers exceptional performance compared to other OpenAI embedding models, especially in multi-language retrieval benchmarks like MIRACL[47]. To store the embeddings, we opted for Chroma, an open-source vector database known for its reliability and efficiency in managing large-scale embedding data. We use a vector store retriever to retrieve the data from the vector store.

## Code executor tool

A code executor tool has been developed to execute Python scripts generated by the AFM Handler Agent to control the AFM software. This tool is intended to run Python code, provided as a text string, directly on the local system to allow for smooth integration with the workflow of the AFM Handler Agent. The utility executes the code and sends back a success message or a detailed description of the error that occurred. If there is an error, the error message is returned to the AFM Handler agent so it can correct the error and retry executing. Otherwise, if the script runs without errors, it is considered the final result. This iterative process ensures precise control of the AFM system while systematically addressing any issues in the script.

## Data handler agent

Surface tracking optimization in AFM requires precise calibration of three fundamental parameters: Proportional ($P$), Integral ($I$), and Derivative ($D$) gains. Optimal calibration manifests as convergence between trace and retrace signals, indicating stable scanning conditions. The Data Handler agent interfaces with specialized optimization and analysis modules; these models can access AFM image data stored in local storage systems. The agent can optimize $P$, $I$, and $D$ gains or calculate various surface properties, such as average friction and surface roughness, using the help of modules and image files stored locally. While many AFM software packages offer basic data analysis

functionalities, they present several limitations in an automated workflow as follows. (i) Most of these software solutions primarily support Windows systems, limiting cross-platform compatibility with operating systems such as macOS and Linux platforms. (ii) Commercial packages require paid licenses, restricting accessibility. (iii) Finally, most packages are not flexible to include additional functionalities beyond what is already included, limiting their customizability. Thus, to ensure broader adaptability and maintain an adaptable, flexible, modular, and open framework, we developed the Data Handler agent within AILA, which has access to several tools—new functionalities can be easily integrated to this agent and the tools based on user needs. Note that this does not restrict the usage of vendor software packages, as they could also be included as a tool in AILA. The agent offers a significantly expanded suite of advanced analytical capabilities, such as:

1. Customizable and automated image processing workflows tailored to specific experimental needs.
2. Statistical analysis across multiple datasets, enabling robust comparison of parameters such as average friction, surface roughness, and topographic variations.
3. Platform independence, ensuring compatibility across Windows, macOS, and Linux, and eliminating reliance on proprietary or licensed software.
4. Dynamic code generation via LLM integration, allowing users to automatically generate and execute scripts for plotting and analysing images.

To demonstrate the adaptability of the agent, we developed a custom function to calculate indentation volume (see Fig. S4 in the Supplementary Information). Instructions for integrating additional functions into the Data Handler Agent are provided in a step-by-step guide available on the accompanying GitHub repository[37].

## Image optimization tool

The feedback system in an AFM plays a crucial role in maintaining control over the interaction between the cantilever tip and the sample surface. During scanning, variations in surface features alter the interaction forces between the tip and the sample, leading to deflections in the cantilever. These deflections are detected by a photodetector. To ensure that these deflections stay within a specified range, the feedback mechanism continuously adjusts the height of either the tip or the sample stage in real-time. This process is managed by a PID controller, which regulates the position of the z-piezo actuator. By controlling the Z position of the AFM probe, the controller maintains a steady interaction force or adheres to a predefined setpoint, depending on the chosen mode of operation.

Fine-tuning the *P*, *I*, and *D* gain values of the controller is vital for achieving accurate control of the setpoint in AFM imaging. The integral gain is especially important for enhancing image clarity by mitigating drift and reducing steady-state errors. Once the integral gain is optimized, adjusting the proportional gain can provide further refinement. The derivative gain, on the other hand, is particularly beneficial for imaging samples with pronounced edge features. If the gains are set too low, the PID loop may fail to maintain the setpoint effectively, while excessively high gain values can introduce electrical noise into the image due to amplified feedback or overcompensation for deviations. Properly optimized PID parameters ensure that the feedback loop remains stable and responsive, enabling the AFM to accurately track surface topography, even at higher scanning speeds. This balance is especially critical when imaging delicate, irregular, or soft materials, as it preserves the integrity of tip-sample interactions.

A genetic algorithm (GA) was employed for PID gain optimization. The GA parameters included a fixed population size of three and a total of 15 generations, enabling efficient tuning of the gains. Although these parameters can be manually adjusted, but excessive image scanning

may degrade the AFM tip. The optimized gains ensure effective feedback control, producing comparable forward and backward images. This can be achieved by calculating the mean squared error (MSE) between forward and backward z-axis images for various PID gain settings. However, this method is sensitive to drift during scanning, and this method also depends on previously acquired images. To address this, the structural similarity index (SSIM) was adopted as the fitness function in the genetic algorithm, providing a robust measure of image similarity between the z-axis forward and backward images, independent of prior image data.

This metric offers advantages over traditional Mean Square Error (MSE) approaches by (i) addressing tip degradation challenges in contact-mode AFM by minimizing required scan cycles and enabling optimization using low-resolution images, (ii) maintaining accuracy under drift conditions, (iii) incorporating structural, brightness, and contrast variations in optimization, and (iv) providing normalized scores between 0 and 1, where 1 indicates perfect similarity.

The SSIM is defined as:

$$SSIM(x,y) = [l(x,y)]^\alpha \times [c(x,y)]^\beta \times [s(x,y)]^\gamma \tag{1}$$

where, $l(x,y)$ is the luminance comparison, $c(x,y)$ is the contrast comparison, and $s(x,y)$ is the structure comparison with $\alpha$, $\beta$, $\gamma$ being the weighting parameters. Note that the individual components are defined as:

$$l(x,y) = (2\mu_x\mu_y + C_1)/(\mu x^2 + \mu y^2 + C_1) \tag{2}$$

$$c(x,y) = (2\sigma_x\sigma_y + C_2)/(\sigma_x{}^2 + \sigma_y{}^2 + C_2) \tag{3}$$

$$s(x,y) = (\sigma_{xy} + C_3)/(\sigma_x\sigma_y + C_3) \tag{4}$$

where, $\mu_x, \mu_y$ represent the mean intensities of images $x$ and $y$, $\sigma_x, \sigma_y$ is the standard deviations of images $x$ and $y$, $\sigma_{xy}$ is the cross-covariance between images $x$ and $y$, and $C_1, C_2, C_3$ are constants to avoid instability with $(C_1 = (k_1L)^2, C_2 = (k_2L)^2, C_3 = C_2/2)$ and $L$ being the dynamic range of pixel values and $k_1 = 0.01$ and $k_2 = 0.03$.

*Baseline correction.* The adaptive baseline correction employed in the step-edge detection of graphene is given by

$$B(x,y) = \Sigma_{i,j}a_{ij}x^iy^j \tag{5}$$

where, $B(x,y)$ is the baseline function, $a_{ij}$ are the polynomial coefficients, $i$ and $j$ are the polynomial degrees ($0 \le i, j \le n$) with $n$ being the maximum polynomial degree.

## Image analyzer tool

AFM instrument stores the image data as a *.nid file in the local system. This *.nid file contains deflection, friction force, and z-axis images for both backward and forward scans. To further process any image from the file, exact data must be extracted from the file. To conduct this, we have used the NSFopen Python library in the Image Analyzer tool, which takes the query from the data handler agent regarding the specific image data and its location and returns the image data in an array to the data handler agent. To conduct further processing of the images, any Python script generated by the data handler agent can be executed in the Image Analyzer tool, and the result can be returned to the data handler agent. Note that there is no database available to guide the LLM model in generating the Python script. It can generate the Python script by itself. There is a total of 6 input parameters for this tool:

1. path (str): directory path to search for the latest file (default: None).
2. Filename (str): specific image file to display (default: None).

(3) Dynamic_code (str): Python code for processing image data (default: None).

(4) Calculate_friction (bool): option to compute average friction (default: False).

(5) Calculate_mean_roughness (bool): option to compute mean roughness (default: False).

(6) Calculate_rms_roughness (bool): option to compute RMS roughness (default: False).

Returns: a dictionary with the status, image data, or error details. Average friction was calculated using the following formula:

$$F_{ave} = \frac{1}{2} \times \left( f_{ij} - b_{ij} \right) \tag{6}$$

where $f_{ij}$ and $b_{ij}$ are the element at position $(i,j)$ in the array of the forward and backward friction image data. We have used the formula in this tool to calculate the mean roughness and RMS roughness values

$$R_{mean} = \frac{1}{M.N} \sum_{i=1}^{M} \sum_{j=1}^{N} \left| z_{ij} - \bar{z} \right| \tag{7}$$

$$R_{rms} = \frac{1}{M.N} \sum_{i=1}^{M} \sum_{j=1}^{N} \left( z_{ij} - \bar{z} \right)^2 \tag{8}$$

where $z_{ij}$ is the element at position $(i,j)$ in the array, $\bar{z}$ is the mean of all elements in the array, $M$ is the number of rows in the array, $N$ is the number of columns in the array of the $z$-axis forward image data.

### Image segment and image scanner tools

An Image Segment and an Image Scanner tool have been created to analyse AFM-scanned images. The Segment tool use the Otsu algorithm to segment images according to the various features present in the sample. Upon detection of features, the tool produces bounding boxes and allocates distinct grain IDs to each feature. This bounding box information is subsequently utilized by LLMs for additional data processing. Note that we used text-based LLM models that cannot discern any features inside the sample, whether they pertain to the material or represent alien inclusions. Following its analysis, the LLM can transmit designated grain IDs to the Image Scanner tool, which instructs the AFM instrument to meticulously scan those particular characteristics.

### AFMBench

**Dataset preparation.** To evaluate the performance of the AILA, we have manually created a set of 100 questions, carefully categorized into three distinct groups. The first classification is based on whether a question requires one or multiple tools and agents to be solved. The second category assesses the complexity of the questions, distinguishing between basic and advanced levels. Lastly, the questions are grouped by their requirements, such as documentation analysis or calculations. The complexity of each question is determined by the number of agents involved and the steps required to achieve the solution. For instance, modifying a parameter in an AFM system typically requires documentation review and the use of a single agent, categorizing it as a basic task. Conversely, capturing an AFM image and analyzing its surface roughness involves multiple agents, documentation analysis, and calculations, making it an advanced task. A comprehensive JSON file has been created, encapsulating detailed metadata about each question, including its respective category, for streamlined analysis and evaluation. This file serves as a structured resource for further investigations and testing. All questions, along with their relevant classifications and details, have been made

accessible on GitHub[37] (https://github.com/M3RG-IITD/AILA) to support transparency and reproducibility in research.

**Evaluation.** We developed a graphical user interface (GUI) using Streamlit, an open-source Python framework, to streamline user interaction with AILA. The GUI allows users to input text-based queries, select the desired LLM model, and specify a log file name. It then executes AILA in the backend, saving the output log file locally and enabling users to observe results directly within the AFM software. Any output images or figures generated by AILA are also stored in the local system for further analysis. To ensure robustness, we manually evaluated all questions using each model, verifying the output log files and AFM software results multiple times in collaboration with different researchers to eliminate potential human errors. The evaluation of AILA's performance was categorized into two metrics: accuracy and efficiency. For accuracy, questions were divided into categories based on complexity and tool or agent usage, with a percentage of correct answers calculated for each category. For efficiency, uniform parameters were maintained across models in the AFM software, including default settings of 0.1 s as time per line and 128 as lines per frame, when not specified by the user. To ensure precise efficiency measurements, scanning time for images and the time taken by questions with incorrect answers were excluded from the analysis. Average response times were computed for each category to assess AILA's overall efficiency.

**Evaluation metrics.** To assess the evaluation of questions in terms of accuracy, we classified the answers provided by AILA into three categories: fully correct answers, incorrect and partially correct answers. A fully correct answer was considered accurate and given a score of 1, while any partially correct response was given a score of 0.5. Given that some questions require manual inspection of the AFM software to verify whether specific parameters are set correctly and whether the AFM image is captured as intended, multiple researchers were involved in verifying the results. They carefully checked the outcomes to ensure error-free results. For measurements of different properties, such as average friction, roughness, and RMS value of roughness, we used the Gwyddion software to verify the accuracy of the results. Subsequently, the questions were clustered into appropriate groups, and the corresponding average percentage of correct answers was calculated. The detailed evaluation process is provided in the Supplementary Information.

Additional evaluation metrics are defined as following. Based on the scoring, Success rate is computed as the average score (out of 100) across three iterations for all the tasks. For all the successful runs, following metrics are employed.

1. AFM handler calls: the average number of times the AFM Handler agent was called to resolve a task (calls per task). An elevated score may signify a greater dependence on the AFM Handler for coordination.

2. Data handler calls: the average number of times the Data Handler agent was called during task solving.

3. Number of steps: the average number of discrete actions (represented by tool or agent call) taken by the AILA to arrive at a solution.

4. Total tokens: total number of LLM model tokens is processed on an average for each task, considering both input (prompt) and output (complete) tokens.

5. Prompt tokens: number of tokens utilized in the conversation's input or instruction segment for each task.

6. Completion tokens: the average number of tokens that the system's answers create for all assignments.

7. Tokens per stage: this tells how many tokens were utilized on average for each stage in the process of solving a task.

8. Latency: the average amount of time (in seconds) it takes to complete a task from the first to the final step.
9. Time per step: the average amount of time (in seconds) spent on each step, representing the pace with which the system can perform different tasks.
10. Latency per 1000 tokens: the average amount of time (in seconds) taken to process 1000 tokens.

**Analysis of prompt structure.** We systematically investigated how prompt structure and phrasing influence GPT-4o's ability to perform load-dependent friction measurements, one of our most challenging open-ended tasks. LLMs process information hierarchically based on input format and length, making them sensitive to prompt design. Even subtle modifications can activate different training exemplars, altering response patterns and reasoning pathways. To evaluate performance within the AILA framework, we constructed multiple input prompts to assess task completion efficacy. In AILA, inter-agent collaboration is triggered when the model outputs "NEED HELP", while "FINAL ANSWER" signals task completion. We developed and tested four distinct prompt categories: (1) concise task descriptions, (2) comprehensive task elaborations, (3) sequential task decompositions, and (4) explicit inclusions of the signaling phrases with case variations. By systematically combining these elements, we designed five system prompts, with detailed performance metrics documented in the Supplementary Information (Table S3).

## Data availability
All the data generated in this study have been deposited in the GitHub repository[37] under accession code https://github.com/M3RG-IITD/AILA. Source data are provided with this paper.

## Code availability
All the codes in AFMBench[37], along with the complete log files of the responses for each of the tasks from all models are available at: https://github.com/M3RG-IITD/AILA.

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

## Acknowledgements

N.M.A.K. acknowledges the funding support from Google Research Scholar Award and the Alexander von Humboldt Foundation. I.M. thanks University Grants Commission (UGC), Government of India for the NET-JRF fellowship (221610021768). J.S. acknowledge the funding received from the Ministry of Education, Government of India, in the form of research fellowship. M.Z. acknowledges the funding received from the PMRF award by the Government of India. M.M.S. acknowledges support from the European Union (ERC, NewGLASS, 101044664). L.W. acknowledges funding from the Carl Zeiss Foundation through its Breakthrough program. The authors thank the IIT Delhi HPC facility for computational and storage resources. We also thank Sushant Sinha for his assistance with the image segmentation.

## Author contributions

I.M. developed the AILA framework, generated the visualizations for the figures and tables, and conducted the experiments. J.S. prepared figures and performed experiments. M.Z. contributed to the code development. M.M.S., K.W., and L.W. edited the draft and validated the results. All authors contributed to the conceptualization of the project, the design of the methodology. I.M. and N.M.A.K. wrote the original draft of the manuscript. All authors contributed to reviewing and editing the manuscript. N.M.A.K. and N.N.G. were responsible for funding acquisition, project administration, resource provision, and overall supervision of the project.

## Competing interests

The authors declare no competing interests.
