## [Transparent Peer Review file · Nature Communications]

Automation of Atomic Force Microscopy with LLM Agents including End-to-End Evaluation Suite

Corresponding Author: Professor N M Anoop Krishnan

Version 0:

Reviewer comments:

Reviewer #1

(Remarks to the Author)

The manuscript presents a framework named AILA, designed to automate microscopy experiments using LLM techniques. The authors evaluate various state-of-the-art LLMs and test AILA's capabilities in handling both microscopy experiments and data analyses. Overall, the study is interesting. Below are some detailed comments:

1. Many AFM operation software or packages already integrate data analysis functionalities. What is the rationale for including a separate data handler tool in AILA? Both AFM experiments and data analysis can be done directly using the same vendor software or package, particularly for the examples provided in this manuscript.
2. What additional or advanced data analysis capabilities does the data handler offer beyond those already existing in AFM vendor software or package?
3. The inclusion of a safety framework is a great development. However, if it is constrained to predefined documentation, it may limit the flexibility for designing experiments not covered in the documentation, particularly for new experiment designs.
4. What is the generic algorithm used for optimizing imaging parameters? Also, if a different prompt or a sample with distinct morphology is used in the experiment, does AILA select a different optimization algorithm?
5. Similar questions, how consistent is the edge detection algorithm applied by AILA? Does the selection of the algorithm depend on real-time results, or is the same method consistently applied across different samples?
6. The consistency and adaptability of algorithm selection might directly impact the reliability of AILA. A more detailed discussion on how AILA ensures robustness in algorithm selection and data analysis should be added.
7. The authors claim that AILA can handle "complex AFM experiments." However, the experiments demonstrated in the manuscript are relatively standard and routine AFM experiments. While the results provide evidence that AI can assist in microscopy operations, the term "complex AFM experiments" is misleading. I recommend the authors rephrase this discussion.
8. Prior research has explored the use of LLMs, such as ChatGPT, for AFM operations. The authors should acknowledge and discuss these efforts, such as <https://iopscience.iop.org/article/10.1088/2632-2153/ad52e9/meta>. Previous work has also identified similar challenges, such as LLMs performing well in single-step experiments but struggling with multi-step processes.
9. In addition, given that this manuscript focuses on LLM-driven autonomous AFM, prior efforts in autonomous AFM development should be properly referenced.

(Remarks on code availability)

Reviewer #2

(Remarks to the Author)

I co-reviewed this with another reviewer. I am not an AFM expert, thus my comments are more focused on the large language model part. I would like to leave AFM part for AFM experts to judge.

The manuscript introduces Artificially Intelligent Lab Assistant (AILA), a framework that automates atomic force microscopy (AFM) tasks through large language model (LLM)-driven agents. They also introduce AFMBench, an evaluation suite designed to challenge AI agents across the full scientific workflow, from experimental design to results analysis. The paper evaluates several state-of-the-art models, including GPT-4o, GPT-3.5-turbo-0125, Llama-3.3-70B-versatile, and Claude-3.5-

sonnet-20241022. The authors also assess AILA's performance in complex AFM experiments, noting prompt fragility where minor changes in prompt structure leads to significant performance variations in models like GPT-4o. The findings stress the importance of robust benchmarking protocols and effective prompt engineering for the successful deployment of AI in scientific research. It is an interesting work, but the following points need to be addressed:

1. This manuscript presents a dynamic routing framework that allows the agent to choose the next agent to use. However, the author didn't discuss how such a dynamic multiagent system actually work and how it performs comparing to single-agent scenario.
2. It is relatively strange to discuss the performance of different LLMs before introducing how it actually solves AFM problems.
3. AFM is an analytical method, and it is used for material analysis to reveal structure-property relationships. However, the authors focused on developing LLM system to improve the autonomy of the AFM analysis, but didn't give real-world examples show how it actually perform AFM analysis, and gives the analysis conclusions like human experts, which is the ultimate goal of such analysis method. It is suggested that authors should add 1-2 real-world examples.
4. In Figure 3 and lines 226-233, the authors discuss the temporal efficiency of various LLMs. Is the task completion time mentioned here the sum of the LLM's response time and the tool execution time? If so, considering that the authors interact with the LLM through network APIs, the network environment of the AILA system and the resources available on the LLM servers could impact the LLM's response time. Therefore, this comparison of results may not accurately reflect the LLM's performance and may need more rigorous handling.
5. The authors mention that Claude-3.5-sonnet exhibits subpar agentic capabilities, primarily due to agent and tool selection errors. This issue may arise from the integration challenges between Anthropic's Claude series models and frameworks like LangChain and LangGraph, especially concerning functionalities like function calls. These integrations might not fully leverage Claude-3.5-sonnet's inherent strengths. I hope the authors can further analyze the reasons for Claude 3.5-sonnet's poor performance from this perspective. Additionally, Anthropic's recent introduction of the Model Context Protocol (MCP) offers a standardized method for connecting AI models with external data sources and tools, potentially enhancing agentic performance. Integrating MCP with Claude has demonstrated improved multi-agent applications, such as Manus. I also hope the authors can conduct research and discussion on this.
6. The authors mention safety alignment in SDLs and implement strict operational boundaries to mitigate safety risks. I suggest that after identifying risky actions, an additional human review mechanism could be introduced. For instance, if an expert detects an issue, they could suggest corrections (input via natural language), enabling a "human-in-the-loop" interaction that ensures continuous, safe operation of the workflow, rather than directly terminating the task when candidate actions beyond operational boundaries.
7. The level of detail and readability of the Supplementary Materials need improvement. (1) A table of contents is missing. (2) The output logs from the AILA system are too lengthy and appear to be directly copied from terminal outputs. These should be organized and refined to enhance readability. (3) The detailed process and results of expert evaluation of AILA's performance on AFMBench, including how researchers determined whether the answers provided by AILA were entirely correct, incorrect, or partially correct, should be included in the supplementary materials and referenced in the main text. If too many details, representative examples can be included.
8. The section titled "Analysis of Prompt Structure" (lines 402-419) in the manuscript appears to be misplaced. This content seems more appropriate for the "Method" section rather than its current location in the manuscript.

(Remarks on code availability)

The provided code gives good descriptions about how it should be used.

Reviewer #3

(Remarks to the Author)

(Remarks on code availability)

Reviewer #4

(Remarks to the Author)

General comments:

Using atomic force microscopy (AFM) as a representative materials characterization tool, the authors present an agentic AI framework based on LLMs for automating the acquisition and analysis of microscopy data. They create a set of

benchmarking tasks and evaluate agents operated by different LLMs, quantifying their completion accuracy, failure modes, and evaluation times. They demonstrate ability of the agent to optimize scan parameters when prompted.

Overall, the manuscript is nicely prepared, with results presented clearly and concisely. The deployment of AI agents for autonomous or assisted experiments has the potential to make a variety of physical techniques more accessible to non-experts. However, experimentally, the scope of the current framework is limited to scans in one location and very standardized analysis. From an applied AI perspective, the agent and tool sets are also highly task-specific and their performance depends greatly on the specificity of the prompts. With the addition of artifact/anomaly detection tools such as those developed by the autonomous microscopy works cited, as well as an additional agent that iteratively debugs generated code to control the instrument, the agentic framework has the potential to be applicable to real-world experiments. Specific questions:

1. In the evaluation of agent performance, is each task given to each model once? Or if the exact same tasks are repeated, how consistent is the agent response each time?
2. The current focus seems to be on the agents completing the task in one go. If an agent cannot complete the task, it knows, with appropriate prompting, to output NEED HELP to invoke the other agent. Is there a way for the user to provide additional context in such situations, and for the user's feedback to be stored to improve the agent?
3. In Table S2, we see detailed vs. concise prompts for the same task. The concise prompt fails but the detailed prompt is a step-by-step guide for the agents. What is the advantage in using an agentic framework here when presumably an LLM with access to tools can follow instructions and execute?
4. Relatedly, the structure of the framework seems to be two agents, which each have access to two tools. How does the performance compare to if one agent was given all tools? Or if there was an agent for each tool?
5. More generally, the distinction between tool and agent is not clear. The authors themselves seem to use the two interchangeably, see for e.g. line 701. Figure 2 and the discussion around it centers on this distinction.
6. Regarding model-specific limitations, could iterative debugging improve results for cases where the failure mode is code generation? I suggest some quantification of the distribution of types of bugs and a clarification of how that may be addressed beyond needing an entirely different LLM.
7. There is a lot of emphasis placed on the model evaluation. At the rate of current progress, models become obsolete in a few months. E.g. Claude 3.5 and GPT-3.5 have been succeeded by newer models.
 - a. It would be nice to see a focus on evaluation metrics rather than specific models
8. One of the critical results presented is the demonstration of experimental workflows beyond single scan acquisition. In figure 5, the prompt given is "Optimize the values of the P, I, and D gains using a genetic algorithm. Do not correct the baseline." This is quite specific. Is there any analysis on whether the agentic framework could complete the task without being told to optimize the PID, but rather more generically with instructions to optimize for image clarity and/or stability between scans? Or without specifying the algorithm to be used?
9. The first three discussion paragraphs summarize and repeat many points that were scattered throughout the results section. I suggest a more in-depth discussion of the last two points instead, as they seem more relevant to addressing strengths (modularity) and limitations (limited functionality) of the framework.
10. AFM parameter optimization:
 - a. Not clear if the genetic algorithm code was provided or written by the LLM. If the former, why not evaluate the models on their ability to write code to solve problems?
 - b. What happens if you add a code writer in addition to a code executor?
11. Does the current workflow call predefined code blocks from the documentation?
12. How can others evaluate their models or agents on this benchmark? By definition, the benchmark should be accessible to anyone.

(Remarks on code availability)

Reviewer #5

(Remarks to the Author)

(Remarks on code availability)

Version 1:

Reviewer comments:

Reviewer #1

(Remarks to the Author)

The authors have addressed my questions.

(Remarks on code availability)

Reviewer #2

(Remarks to the Author)

I believe the quality of the manuscript has improved substantially after the initial review.

However, I still have two minor concerns that I would like to address:

Q1: On lines 279-280, the manuscript mentions that single-agent architectures offer computational advantages through reduced intermediate processing and lower latency. However, there is no specific data or evidence provided to support this claim.

Q2: On line 271, the manuscript introduces the concepts of single-agent and multi-agent systems. However, in the subsequent sections, particularly lines 277-292, there seems to be some confusion regarding whether the same concepts are being discussed. Additionally, the data in Figure 3d and Figure S5 appear to be inconsistent, which suggests that these two sections might be referring to different concepts under the same terminology. Could the authors please clarify the relationship between these concepts and ensure consistency in the terminology used throughout the manuscript?

(Remarks on code availability)

The code is well-documented.

Reviewer #3

(Remarks to the Author)

(Remarks on code availability)

Reviewer #4

(Remarks to the Author)

The authors have provided a detailed and comprehensive response to our review comments. I have no further questions or comments.

(Remarks on code availability)

Reviewer #5

(Remarks to the Author)

(Remarks on code availability)

Dear Editor and Reviewers,

Thank you for considering our manuscript and for providing valuable feedback. We have carefully addressed all the comments raised by the reviewers and editor by performing several additional experiments. Corresponding changes are incorporated in the revised versions of supplementary file or the main manuscript (with the changes highlighted in blue colour).

Below, we provide a detailed point-by-point response to each reviewer comment. We believe these revisions have significantly strengthened the manuscript and hope that you now find it suitable for publication.

REVIEWER COMMENTS

Reviewer #1 (Remarks to the Author):

The manuscript presents a framework named AILA, designed to automate microscopy experiments using LLM techniques. The authors evaluate various state-of-the-art LLMs and test AILA's capabilities in handling both microscopy experiments and data analyses. Overall, the study is interesting. Below are some detailed comments:

Response: Thank you for your positive feedback on our manuscript. We sincerely appreciate your valuable insights, which have helped us enhance the quality of the paper. Thank you again for your time and support.

1. Many AFM operation software or packages already integrate data analysis functionalities. What is the rationale for including a separate data handler tool in AILA? Both AFM experiments and data analysis can be done directly using the same vendor software or package, particularly for the examples provided in this manuscript.

Response: Thank you for raising this point. We agree that most AFM operation software or packages integrate some level of data analysis functionalities; however, these are typically limited to basic tasks such as viewing images or measuring feature sizes. Although a few AFM brand-specific software packages offer more extended functionalities, they are rarely capable of performing advanced-level data analysis—such as image processing, calculating average friction, and extracting other complex properties. For such applications, researchers often rely on specialized software packages such as Gwyddion (open source) or SPIP (commercial).

In fact, before integrating the Data Handler agent into AILA, we attempted to incorporate such existing software packages within our framework. However, several challenges emerged from an implementation stand-point: (i) most of these software solutions primarily support Windows systems, limiting cross platform compatibility with operating systems such as macOS and Linux platforms; (ii) some require paid licenses, restricting accessibility; and finally, (iii) the packages are not flexible to include additional functionalities beyond what is already included limiting

customizability. Thus, to ensure broader adaptability, and maintain an adaptable, flexible, modular and open framework, we developed the Data Handler agent, which has access to several tools, within AILA—new functionalities can be easily integrated to this agent and the tools based on user needs. Note that this does not restrict the usage of vendor software packages as they could also be included as a tool in AILA.

To address this comment, we have added the following text to the **Methodology section (Data Handler Agent subsection)** of the main manuscript.

“While many AFM software packages offer basic data analysis functionalities, they present several limitations in an automated workflow as follows. (i) Most of these software solutions primarily support Windows systems, limiting cross platform compatibility with operating systems such as macOS and Linux platforms. (ii) Commercial packages require paid licenses, restricting accessibility. (iii) Finally, most packages are not flexible to include additional functionalities beyond what is already included, limiting their customizability. Thus, to ensure broader adaptability, and maintain an adaptable, flexible, modular and open framework, we developed the Data Handler agent within AILA, which has access to several tools—new functionalities can be easily integrated to this agent and the tools based on user needs. Note that this does not restrict the usage of vendor software packages as they could also be included as a tool in AILA.”

2. What additional or advanced data analysis capabilities does the data handler offer beyond those already existing in AFM vendor software or package?

Response: As mentioned earlier, while traditional AFM vendor software typically provides basic functionalities such as image visualization, simple feature measurements, and limited data processing, the Data Handler agent integrated into AILA offers a significantly expanded suite of advanced analytical capabilities. These include:

1. Customizable and automated image processing workflows tailored to specific experimental needs.
2. Statistical analysis across multiple datasets, enabling robust comparison of parameters such as average friction, surface roughness, and topographic variations.
3. Platform independence, ensuring compatibility across Windows, macOS, and Linux, and eliminating reliance on proprietary or licensed software.
4. Dynamic code generation via LLM integration, allowing users to automatically generate and execute scripts for plotting and analysing images.

To further address the Reviewer’s question and demonstrate such extensibility of the framework, we have integrated a new function in the Image Analyzer tool of the Data Handler agent. This function calculates the indentation volume (or wear volume) using topographic AFM data as below. Figure S4 shows the input and output of this new function. These results demonstrate how a new function or tool can be seamlessly implemented in the AILA framework.

The implemented code snippet is as follows.

```
Z = data['Image']['Forward']['Z-Axis'] * 1e9 # read data
# Initial flattening
Z_flat = flatten(Z, order=1)
```

```

# Extract spatial extents in micrometers (µm)
param = afm.param
x_min = param['X']['min'][0] * 1e6
x_max = x_min + param['X']['range'][0] * 1e6
y_min = param['Y']['min'][0] * 1e6
y_max = y_min + param['Y']['range'][0] * 1e6
extents = [x_min, x_max, y_min, y_max]

# Plot flattened image
clip = [np.percentile(Z_flat, percent) for percent in [0, 100]]

# Threshold for detecting indentation (in nanometers)
threshold = -5 # nm
mask = Z_flat < threshold

# Estimate base height from surrounding area
base_height = np.mean(Z_flat[~mask])

# Calculate depth map (in nanometers)
depth_map = base_height - Z_flat
depth_map[depth_map < 0] = 0
depth_map[~mask] = 0 # keep only indentation region

# Convert depth map to micrometers
depth_map_um = depth_map * 1e-3 # nm → µm

# Pixel area calculation (in µm²)
pixel_size_x = (x_max - x_min) / Z_flat.shape[1]
pixel_size_y = (y_max - y_min) / Z_flat.shape[0]
pixel_area = pixel_size_x * pixel_size_y # µm²

# Volume = sum of (depth × area) → µm³
volume = np.sum(depth_map_um) * pixel_area

print(f"Auto-detected indentation volume: {volume:.4f} µm³")
from datetime import datetime
timestamp = datetime.now().strftime("%Y%m%d_%H%M%S")
filename = f"indentation_depth_map_{timestamp}.png"
plt.imshow(depth_map, cmap='viridis', extent=extents, origin='lower')
plt.colorbar(label="Depth (nm)")
plt.title("Indentation Depth Map")
plt.xlabel("X (µm)")
plt.ylabel("Y (µm)")
plt.savefig(filename, dpi=300, bbox_inches='tight')

```

Based on the above, we have included the following text to the **Methodology section (Data Handler Agent subsection)** of the main manuscript and added the code snippet and the new Figure S4 in the supplementary material (Section S3).

“The agent offers a significantly expanded suite of advanced analytical capabilities such as:

1. *Customizable and automated image processing workflows tailored to specific experimental needs.*
2. *Statistical analysis across multiple datasets, enabling robust comparison of parameters such as average friction, surface roughness, and topographic variations.*
3. *Platform independence, ensuring compatibility across Windows, macOS, and Linux, and eliminating reliance on proprietary or licensed software.*
4. *Dynamic code generation via LLM integration, allowing users to automatically generate and execute scripts for plotting and analysing images.*

To demonstrate the adaptability of the tool, we developed a custom function to calculate indentation volume (see Figure S4). Instructions for integrating additional functions into the Data Handler Agent are provided in a step-by-step guide available on the accompanying GitHub repository.”

Figure S4. Output of a custom indentation volume calculation function within the Data Handler Agent.

3. The inclusion of a safety framework is a great development. However, if it is constrained to predefined documentation, it may limit the flexibility for designing experiments not covered in the documentation, particularly for new experiment designs.

Response: Thank you for raising this important point. We classified all the operations that could be performed on an AFM as per the instrument's documentation into two categories: (i) General operations—these include setting imaging parameters, controlling the tip, selecting the scanning area, and other standard tasks; and (ii) Critical operations—these involve sensitive adjustments such as factory calibrations, laser alignment, piezo calibration, and thermal calibration. Even a minor coding error in critical operations could seriously damage the instrument.

In AILA, we restricted the access to the documentation associated with critical operations, while allowing complete access to the documentation of general operations. For now, the critical functions are limited to trained human experts. Note that general operations were selected in such a fashion that any experiment, that could potentially be performed on an AFM, could be carried out using some combination of these operations. Thus, the predefined documentation in AILA does not restrict any new experiment to be performed on the AFM instrument. The safety frameworks are implemented to prevent actions that may be detrimental to the equipment.

That said, as mentioned earlier, the AILA framework is designed to be flexible and can be easily updated to support access to critical operations as well. In this case, reward functions that prevent unsafe usages could be implemented to ensure safety while performing experiment. This approach is beyond the scope of the present manuscript and is being explored as part of future work.

To clarify this, we have added additional text in **Section 2.5** of the main manuscript, clarifying general operations and critical operations as follows.

“To implement this, we classified all the operations that could be performed on an AFM as per the instrument’s documentation into two categories. (i) General operations—these include setting imaging parameters, controlling the tip, selecting the scanning area, and other standard tasks. (ii) Critical operations—these involve sensitive adjustments such as factory calibrations, laser alignment, piezo calibration, and thermal calibration. Even a minor coding error in critical operations could seriously damage the instrument.”

We restricted the access to the documentation of critical operations in AILA, while allowing complete access to the documentation of general operations. The critical functions are limited to trained human experts. Note that general operations were selected in such a fashion that any experiment that could potentially be performed on an AFM could be carried out using some combination of these operations. Thus, the predefined documentation in AILA does not restrict any new experiment to be performed on the AFM instrument.”

4. What is the generic algorithm used for optimizing imaging parameters? Also, if a different prompt or a sample with distinct morphology is used in the experiment, does AILA select a different optimization algorithm?

Response: The generic algorithm we used for optimizing imaging parameters is NSGA-II (Non-dominated Sorting Genetic Algorithm II).¹ It is a widely used multi-objective evolutionary algorithm that efficiently handles trade-offs between competing objectives, making it well-suited for complex imaging optimization problems.

To generate optimized images, AILA goes through two key processes: 1) baseline correction, and 2) optimization using NSGA-II. Baseline correction is crucial because it accounts for variations in sample morphology. AILA first analyses the input and decides whether baseline correction is necessary or not, based on the distinct morphology of the investigated sample. If needed, it adjusts the baseline accordingly to ensure accurate optimization.

Afterwards, AILA proceeds to the optimization step. While several algorithms could be used for this optimization, we chose NSGA-II based on its robustness, flexibility, and ability to efficiently explore a large parameter space, even in complex and non-linear problems. This matches the diverse data AILA handles. When a different prompt or a sample with distinct morphology is used, AILA dynamically decides whether baseline correction is necessary or not.

To further demonstrate this, we tested two different prompts with samples of distinct morphologies, as shown in the figure below. For the first case, where the sample had small-sized features (as mentioned in the prompt by user), AILA chose to apply baseline correction, recognizing that such features typically require correction to improve signal clarity (baseline: True). In the second case, where baseline correction was not essential due to the sample morphology, AILA skipped the correction step (baseline: False). Thus, in both cases, AILA correctly selected the appropriate agent and tool.

We have added the following text in **Section 2.6.2** of the manuscript and added a new Figure S3 in Supplementary Information.

“To further demonstrate this, we tested two different prompts with samples of distinct morphologies (see Figure S3 in Supplementary Information). In both cases, AILA correctly selected the appropriate baseline correction.”

Figure S3. Demonstration of AILA’s baseline correction decisions for distinct morphologies. For the first case, where the sample had small-sized features (as mentioned in the prompt by user), AILA chose to apply baseline correction, recognizing that such features typically require correction to improve signal clarity (baseline: True). In the second case, where baseline correction was not essential due to the sample morphology, AILA skipped the correction step (baseline: False).

5. Similar questions, how consistent is the edge detection algorithm applied by AILA? Does the selection of the algorithm depend on real-time results, or is the same method consistently applied across different samples?

Response: In the AILA framework, edge detection is not handled by a fixed, hardcoded algorithm. Instead, the system is designed with flexibility at its core. While some image processing tasks may use predefined algorithms with tuneable parameters (these can be adjusted by LLMs based on user input), edge detection specifically is treated as a dynamic task.

To this end, AILA generates new code on-the-fly to solve the edge detection problem based on the context and characteristics of each image. This ensures that the method is adaptable and optimized for different samples rather than relying on a one-size-fits-all approach. Furthermore, we have now updated AILA to incorporate new tools for feature detection, namely, Image Segment and Image Scanner tools. The Image Segment tool is based on Otsu’s thresholding algorithm. This tool automatically segments images by identifying an optimal threshold that separates the foreground from the background or partitions the image into multiple regions based on intensity levels. By maximizing the between-class variance, the Otsu method ensures effective segmentation, which enhances the accuracy and reliability of downstream feature extraction and analysis.

We have added the following text in **Section 2.6.2** of the revised manuscript.

“Note that in the AILA framework, edge detection is not based on a fixed algorithm. Instead, the system generates custom code to solve the problem whereas for feature detection, AILA uses the built-in Image Segment tool (see Methodology) that applies Otsu’s thresholding to automatically segment images by finding the most effective intensity-based thresholds.”

Additionally, we have added the description of the tools in the **Methodology** section of the manuscript as follows.

“Image Segment and Image Scanner tools

An Image Segment and an Image Scanner tool have been created to analyse AFM-scanned images. The Segment tool use the Otsu algorithm to segment images according to the various features present in the sample. Upon detection of features, the tool produces bounding boxes and allocates distinct grain IDs to each feature. This bounding box information is subsequently utilized by LLMs for additional data processing. Note that we used text-based LLM models that cannot discern any features inside the sample, whether they pertain to the material or represent alien inclusions. Following its analysis, the LLM can transmit designated grain IDs to the Image Scanner tool, which instructs the AFM instrument to meticulously scan those particular characteristics.”

6. The consistency and adaptability of algorithm selection might directly impact the reliability of AILA. A more detailed discussion on how AILA ensures robustness in algorithm selection and data analysis should be added.

Response: AILA incorporates two distinct approaches for algorithm selection, each designed to balance robustness with flexibility. In some scenarios, AILA employs pre-defined (hardcoded) algorithms where only the input parameters are adjusted dynamically based on the user’s query. This ensures a high degree of consistency and reliability, especially for standard calculations such as average friction, roughness, etc., where reproducibility is crucial. On the other hand, in more exploratory or visualization-focused tasks, such as edge detection or generating plots, AILA dynamically writes the necessary code. This method introduces a high degree of flexibility, allowing AILA to tailor the analysis more closely to specific data structures or user requirements. While this adaptive capability enhances versatility, we acknowledge that it can also introduce variability in results depending on the complexity of the task and the data involved.

We have added the following paragraph in the **Methodology Section (AILA subsection)** of the manuscript.

“AILA uses two different approaches for selecting algorithms, depending on the task. When performing standard calculations like friction or roughness analysis, AILA relies on established algorithms with adjustable input parameters. This method produces consistent and reproducible results. For exploratory work and data visualization, AILA takes a different approach by creating the code on-the-fly. This method adapts better to varying data formats and specific user requirements; however this may introduce variability in the results depending on the complexity of the task.”

7. The authors claim that AILA can handle "complex AFM experiments." However, the experiments demonstrated in the manuscript are relatively standard and routine AFM experiments. While the results provide evidence that AI can assist in microscopy operations, the term "complex AFM experiments" is misleading. I recommend the authors rephrase this discussion.

Response: Thank you for your thoughtful feedback. We appreciate your point regarding the use of the term "complex AFM experiments." We agree that the examples provided in the manuscript are mostly considered standard within the AFM community and that the aim of the manuscript was to demonstrate that AI agents can automate microscopy operations. To address your concern and avoid any potential confusion, we have removed the term "complex" in the revised version of the manuscript.

To further address the Reviewer's comment, and to explore the frontier capabilities of AILA, we have now performed two additional real-world experiments in the revised version of the manuscript. The first experiment involves identifying a graphene flake on a silicon wafer and estimating the number of graphene layers present. The second experiment focuses on detecting an indentation mark and determining which indenter was used in the process. We believe these experiments represent relatively complex tasks within the domain of AI-assisted microscopy, particularly when performed without human intervention.

Please refer to Point 3 of Reviewer #2 for more details about the experiments, results, and the corresponding changes in the manuscript.

8. Prior research has explored the use of LLMs, such as ChatGPT, for AFM operations. The authors should acknowledge and discuss these efforts, such as <https://iopscience.iop.org/article/10.1088/2632-2153/ad52e9/meta>. Previous work has also identified similar challenges, such as LLMs performing well in single-step experiments but struggling with multi-step processes.

Response: We have added the suggested reference and included the following line in the revised manuscript:

"In addition to these targeted advancements, Liu et al. explores the integration of large language models with APIs to enhance workflow preparation, instrument operation, and data reproducibility in scanning probe microscopy research."

9. In addition, given that this manuscript focuses on LLM-driven autonomous AFM, prior efforts in autonomous AFM development should be properly referenced.

Response: We have added references to prior efforts in autonomous AFM development to appropriately acknowledge the existing work.

Following new references have been added in the revised manuscript.

"1. Liu, Y. et al. Learning the right channel in multimodal imaging: automated experiment in piezoresponse force microscopy. npj Computational Materials 9, 34 (2023).

2. Liu, Y. et al. *Disentangling Electronic Transport and Hysteresis at Individual Grain Boundaries in Hybrid Perovskites via Automated Scanning Probe Microscopy*. *ACS Nano* 17, 9647–9657 (2023).

3. Liu, Y., Checa, M. & Vasudevan, R. K. *Synergizing human expertise and AI efficiency with language model for microscopy operation and automated experiment design*. *Machine Learning: Science and Technology* 5, 02LT01 (2024)”

Reviewer #2 (Remarks to the Author):

I co-reviewed this with another reviewer. I am not an AFM expert, thus my comments are more focused on the large language model part. I would like to leave AFM part for AFM experts to judge.

The manuscript introduces Artificially Intelligent Lab Assistant (AILA), a framework that automates atomic force microscopy (AFM) tasks through large language model (LLM)-driven agents. They also introduce AFMBench, an evaluation suite designed to challenge AI agents across the full scientific workflow, from experimental design to results analysis. The paper evaluates several state-of-the-art models, including GPT-4o, GPT-3.5-turbo-0125, Llama-3.3-70B-versatile, and Claude-3.5-sonnet-20241022. The authors also assess AILA’s performance in complex AFM experiments, noting prompt fragility where minor changes in prompt structure leads to significant performance variations in models like GPT-4o. The findings stress the importance of robust benchmarking protocols and effective prompt engineering for the successful deployment of AI in scientific research. It is an interesting work, but the following points need to be addressed:

Response: Thank you for your careful review and for highlighting the strengths of our work. We sincerely appreciate your constructive feedback and the time you invested in evaluating our manuscript. We have addressed all of your concerns point-by-point in our detailed responses below.

1. This manuscript presents a dynamic routing framework that allows the agent to choose the next agent to use. However, the author didn't discuss how such a dynamic multiagent system actually work and how it performs comparing to single-agent scenario.

Response: The AILA framework incorporates two key functions that enable dynamic routing between agents. Upon receiving a user query, AILA leverages a large language model (LLM) guided by a structured system prompt to determine the most suitable agent to handle the task. The selected agent’s name is embedded in the model’s response. The first routing function parses the last message to extract the agent identifier and subsequently dispatches the query to the corresponding agent using a function call. This mechanism empowers AILA to dynamically select and activate agents based on the nature of the input. The second function handles agent-to-agent task delegation. It checks whether the currently active agent has made any tool calls. If not, the function scans the most recent message for a “NEED HELP” flag. Upon detecting this, it invokes another routing function, transferring the unresolved task along with all previous context to the next agent in line. This iterative process

continues until an agent returns a message containing the termination condition “FINAL ANSWER”. In contrast, for static routing scenarios, AILA enforces a strict policy where the output from a tool call is mandatorily passed back to the invoking agent. This ensures a deterministic, unidirectional data flow from tools to agents, thereby preserving all intermediate outputs and preventing data loss.

To address this comment, following new text have been added in the **Section 2.1** of the main manuscript. Additional detailed explanations have now been provided in **Section S2.4** of the revised Supplementary Information with reference to this section in the main manuscript.

“This planner directly takes query from a user and identifies the appropriate agent to handle the task. The agent makes tool calls to carry out the necessary steps to complete the experiment. The agent-to-agent coordination is invoked by two keywords, namely, “NEED HELP” and “FINAL ANSWER”. While the former one invokes a routing function that transfers the unresolved task to the next appropriate agent, the latter results in the termination of the experiment. Thus, AILA employs a dynamic, routing exploiting available agents and tools, for completing the task given by the user (see S2.4 in Supplementary Information for additional details).”

2. It is relatively strange to discuss the performance of different LLMs before introducing how it actually solves AFM problems.

Response: Thank you for your valuable feedback. While the demonstration of how AILA solves AFM problems was included briefly in the original manuscript (see Fig. 1), in addition to the explanation above (to Point 1), we have now included additional clarification and explanation in the updated manuscript, specifically in Figure 1c. This figure outlines the full problem-solving pipeline, providing the necessary context before performance comparisons.

Furthermore, to improve clarity and further address your concern, we have made the following changes in **Section 2.1** of the revised manuscript.

“In an AFM experiment, the workflow usually involves two key steps: capturing the image and analysing the results. The imaging part starts with choosing the right cantilever, then setting the imaging parameters. Afterwards, the tip is gently moved toward the sample surface, and the scan is carried out. For every stage, AILA creates a specific Python script and executes it, controlling the AFM instrument in real time through an API. This connection allows the digital commands to directly translate into physical movements on the instrument. Once the scan is complete, the image is saved automatically and opened for analysis.”

3. AFM is an analytical method, and it is used for material analysis to reveal structure-property relationships. However, the authors focused on developing LLM system to improve the autonomy of the AFM analysis, but didn't give real-world examples show how it actually perform AFM analysis, and gives the analysis conclusions like human experts, which is the ultimate goal of such analysis method. It is suggested that authors should add 1-2 real-world examples.

Response: As mentioned in our response to Point 7 of Reviewer #1 above, we have now added two experiments that AFM experts identify as representative of real-world

example. Further, we demonstrate that AILA can provide analysis and conclusions similar to human experts.

To this end, we consider two examples. (i) Identifying a graphene flake on a silicon wafer and estimating the number of graphene layers present. (ii) Detecting an indentation mark and determining which indenter was used in the process. Figure 6b,c below shows the prompt, corresponding images, and the analysis provided by AILA. Interestingly, in both the cases, AILA performs the experiment successfully and provides analysis and conclusions similar to human experts. Specifically, in the first case, it uses the knowledge of the thickness of graphene along with numerical computation to identify the number of layers, whereas in the second, it uses the knowledge on the characteristics of Vickers and conosphical indenters along with reason to identify the appropriate indenter.

To address this comment, **Figure 6** in the main manuscript has been modified to include the new subfigures b,c (see below) and related text in **Section 2.6** has been modified in the revised version of the manuscript as follows.

Figure 6. Real-world AFM experiments and analysis. b, Demonstration of AILA's workflow for real-world experimentation on graphene-coated Si sample, displaying the user's transcribed query, AILA's unedited final response, intermediate analyses, and exported images. This showcases AILA's capability to autonomously conduct real-

world experimental tasks. c, Demonstration of how AILA identifies an indentation mark on a glass substrate, analyses the indenter type using a horizontal line profile, and provides a final interpretation with supporting explanation.

To evaluate the performance of AILA in a real-world experimental setting, we designed two distinct experiments. In both cases, an experimentalist is required to identify a specific feature of interest, capture it, and then carry out the experiment. Further, the results need to be analyzed to draw meaningful conclusions. In the first experiment (see Figure 6b), the objective is to locate a graphene flake and determine the number of atomic layers in it. To accomplish this, AILA performs image segmentation using Image Segment tool within a user-specified region, identifies the largest visible flake, and extracts it for further analysis. Through a sequence of intermediate processing steps, AILA autonomously generates code, processes the image, and ultimately provides an estimate of the number of graphene layers present in the selected flake. The second experiment involves identifying the type of indenter used to create an impression on a sample surface. The intended image is analyzed by AILA, including an inspection of the indentation line profile. Based on this analysis, it infers and concludes, along with a detailed explanation, that the indenter used was most likely of Vickers-type geometry. Thus, in both the cases, AILA performs the experiment successfully and provides analysis and conclusions similar to human experts. Specifically, in the first case, it uses the knowledge of the thickness of graphene along with numerical computation to identify the number of layers, whereas in the second, it uses the knowledge on the characteristics of Vickers and conospherical indenters along with reason to identify the appropriate indenter. The complete set of raw user inputs and unprocessed outputs are presented in Figures 6b, c. The log file in both experiments is publicly available in our GitHub repository.

4. In Figure 3 and lines 226-233, the authors discuss the temporal efficiency of various LLMs. Is the task completion time mentioned here the sum of the LLM's response time and the tool execution time? If so, considering that the authors interact with the LLM through network APIs, the network environment of the AILA system and the resources available on the LLM servers could impact the LLM's response time. Therefore, this comparison of results may not accurately reflect the LLM's performance and may need more rigorous handling.

Response: Yes, the task completion times presented include both the LLM's response time and the subsequent tool execution time. Our intention in reporting this combined metric was to evaluate the end-to-end latency of task completion within the AILA system, as perceived by the end user. This includes all relevant delays—model inference, network communication, and backend processing—which are critical factors in real-world usability, especially for interactive scientific workflows.

To mitigate variability, all API calls were made under controlled and consistent network conditions, and the tests were repeated multiple times to reduce the impact of transient fluctuations. We acknowledge that network conditions and server resource availability can introduce variability. However, since our primary focus was on comparing relative latency across different LLMs under similar usage conditions within the same framework, we believe the results still provide meaningful insight into how different models perform in practice. Additionally, by consistently measuring the full interaction loop, we aim to highlight practical considerations that users must take into account

when integrating LLMs into systems like AILA. Furthermore, we acknowledge that presenting temporal efficiency across different task distributions and module utilizations in AFMbench may cause confusion. Therefore, we have removed this detailed breakdown from the main manuscript and now **report only the average latency** for clarity and conciseness. Note that Figure 3 has now been updated to include several additional metrics, which are now discussed in detail in the main manuscript.

5. The authors mention that Claude-3.5-sonnet exhibits subpar agentic capabilities, primarily due to agent and tool selection errors. This issue may arise from the integration challenges between Anthropic's Claude series models and frameworks like LangChain and LangGraph, especially concerning functionalities like function calls. These integrations might not fully leverage Claude-3.5-sonnet's inherent strengths. I hope the authors can further analyze the reasons for Claude 3.5-sonnet's poor performance from this perspective.

Additionally, Anthropic's recent introduction of the Model Context Protocol (MCP) offers a standardized method for connecting AI models with external data sources and tools, potentially enhancing agentic performance. Integrating MCP with Claude has demonstrated improved multi-agent applications, such as Manus. I also hope the authors can conduct research and discussion on this.

Response: We agree that MCP provides a standardized and effective method for interfacing large language models with external tools and data sources. To investigate the potential benefits of MCP, we selected a representative subset of 10 diverse questions from the 100 questions in the AFMBench dataset, ensuring coverage of various task types. We conducted evaluations using both the LangGraph framework and MCP-based integration. The comparative results from both approaches are presented below (see Table S2, revealing that the performance metrics were largely similar across the two frameworks).

One of the commonly reported limitations with LLM-based agents, particularly in LangChain-like environments, is inaccurate or failed function calls. However, in our case, both LangGraph and MCP consistently invoked the correct functions and executed the selected tools as expected. Therefore, tool invocation errors were not the primary cause of degraded performance.

Upon deeper analysis, we identified that the core issue with Claude 3.5-sonnet's performance lies in its poor reasoning capabilities, specifically in agent or tool selection. Despite being provided with the same system prompt and function set, the model frequently selected suboptimal or entirely incorrect tools to address the given tasks. This misalignment was not due to a failure in tool execution but rather an incorrect decision at the reasoning stage. To illustrate this, we have included selected output logs below. These examples clearly show instances where the model choose an inappropriate agent or tool despite all required metadata and documentation being present. The issue appears to stem from a confusion in tool differentiation when multiple tools are semantically or functionally similar. It further clarifies that while integration protocols are crucial, reasoning and decision-making within the model remain key bottlenecks.

Protocol	Accuracy
MCP	37%
LangGraph	43%

Table S2: Accuracy Evaluation of Claude 3.5 Sonnet Across MCP and LangGraph Protocols.

```

('user', "Open the file 'Sample.nid', extract the data corresponding to the Z-axis forward image, calculate the
number of square grids present in the image, and save the Z-axis forward image as 'question_22.png'")
=====
                                                    Ai                Message
=====
Name: AILA

Let me help coordinate this task. This request involves multiple steps:

1. Opening and extracting data from an AFM file (.nid)
2. Processing the Z-axis forward image data
3. Analyzing the image to count square grids
4. Saving the image

First, we need the AFM_Handler to open the file and extract the Z-axis forward image data.

I'll select: AFM_Handler
=====
                                                    Ai                Message
=====
Name: AFM_Handler

```

We have added the following text in **Section 2.3** of the revised manuscript. Further, we included detailed discussion of the results in the Section S3.4 of the Supplementary Information.

“To further investigate whether the poor performance originates from the LangGraph framework, we implemented the Model Context Protocol (MCP) to assess the performance of Claude (see Section S3.4 in the Supplementary Information for detailed results). We observed the results from both the frameworks were consistent confirming that the diminished performance is inherent to the model and not a result of the framework.”

6. The authors mention safety alignment in SDLs and implement strict operational boundaries to mitigate safety risks. I suggest that after identifying risky actions, an additional human review mechanism could be introduced. For instance, if an expert detects an issue, they could suggest corrections (input via natural language), enabling a "human-in-the-loop" interaction that ensures continuous, safe operation of the workflow, rather than directly terminating the task when candidate actions beyond operational boundaries.

Response: In the current study, the framework has been deliberately designed to evaluate whether the system, driven by an LLM, can autonomously determine its capability to solve a given task and, when necessary, initiate the invocation of additional agents without any human intervention. This includes supporting multiple iterations of agent invocation until a solution is found or the process concludes based on predefined iteration constraints. Our goal was to rigorously assess the self-sufficiency and adaptability of the agent network under conditions that assume no

external input or feedback. While incorporating user feedback mechanisms is indeed a promising and valuable direction, it falls outside the scope of this work. We are actively exploring new frameworks that incorporate user interaction and adaptive feedback as part of our future research. We truly appreciate your insightful feedback and hope this response helps clarify the design choices and objectives of the current implementation.

To address this comment, we added the following new text to Section 2.5 of the main manuscript.

“Note that an alternate approach to implement safety is to identify risky actions and involve a human-in-the-loop mechanism to review the actions. This could also enhance the robustness and overall performance of the framework. However, such a framework requires human supervision limiting the high-throughput nature that an autonomous system can otherwise achieve, with human response becoming the bottleneck. Hence, this approach was not implemented in AILA, and could be explored as part of future work.”

7. The level of detail and readability of the Supplementary Materials need improvement. (1) A table of contents is missing. (2) The output logs from the AILA system are too lengthy and appear to be directly copied from terminal outputs. These should be organized and refined to enhance readability. (3) The detailed process and results of expert evaluation of AILA’s performance on AFMBench, including how researchers determined whether the answers provided by AILA were entirely correct, incorrect, or partially correct, should be included in the supplementary materials and referenced in the main text. If too many details, representative examples can be included.

Response: We have revised the Supplementary Information accordingly: (1) a table of contents with hyperlinks has been added for easier navigation; (2) system output logs have been cleaned and reformatted to remove raw terminal dumps and improve readability; (3) a detailed description of the expert evaluation process on AFMBench, including the criteria for labelling responses as correct, partially correct, or incorrect, has been included in **Section S4** of the revised Supplementary information, along with representative examples. Please see the updated Supplementary Information file.

8. The section titled “Analysis of Prompt Structure” (lines 402-419) in the manuscript appears to be misplaced. This content seems more appropriate for the “Method” section rather than its current location in the manuscript.

Response: Thank you for the careful reading. We have moved the “Analysis of Prompt Structure” section to the “Methodology” section as recommended to improve the logical flow and organization of the manuscript.

However, since the results of the prompt analysis was used to design further prompts and also form a key result, the observations from the prompt analysis was included in **Section 2.6.3** of the main manuscript.

“However, based on the experiments conducted above and this experiment, we observed that the performance of the LLMs could be directly affected by the prompts. To evaluate this effect, we analyzed the effect of prompting (see Methodology and Table S3) by systematically varying the prompts from simple to complex, from compact

to descriptive. Our findings revealed that GPT-4o demonstrated variable task completion rates across different prompt structures, ranging from partial execution to complete fulfillment. Significantly, more elaborate and detailed prompts consistently enhanced the model's performance reliability and execution accuracy, suggesting that comprehensive contextual information improves complex task handling. Based on these experiments, we designed the prompts consistently across the experiments and ensured that prompt optimization was not carried out to arrive at desirable results."

Reviewer #2 (Remarks on code availability):

The provided code gives good descriptions about how it should be used.

Response: Thank you for your positive feedback regarding the provided code.

Reviewer #3 (Remarks to the Author):

Response: Thank you for taking the time to co-review our manuscript and for contributing to the detailed feedback. We truly appreciate your effort and collaboration in the review process.

Reviewer #4 (Remarks to the Author):

General comments:

Using atomic force microscopy (AFM) as a representative materials characterization tool, the authors present an agentic AI framework based on LLMs for automating the acquisition and analysis of microscopy data. They create a set of benchmarking tasks and evaluate agents operated by different LLMs, quantifying their completion accuracy, failure modes, and evaluation times. They demonstrate ability of the agent to optimize scan parameters when prompted.

Overall, the manuscript is nicely prepared, with results presented clearly and concisely. The deployment of AI agents for autonomous or assisted experiments has the potential to make a variety of physical techniques more accessible to non-experts. However, experimentally, the scope of the current framework is limited to scans in one location and very standardized analysis. From an applied AI perspective, the agent and tool sets are also highly task-specific and their performance depends greatly on the specificity of the prompts. With the addition of artifact/anomaly detection tools such as those developed by the autonomous microscopy works cited, as well as an additional agent that iteratively debugs generated code to control the instrument, the agentic framework has the potential to be applicable to real-world experiments.

Response: Thank you for the positive feedback. Please note the following points regarding the scope of the framework and agent and tool capabilities.

1. We would like to clarify that the current framework is not limited to scans in a single location—AILA can perform multiple scans and synthesize information across scan from various locations (see **Point 7 to Reviewer #1 and Point 3 to Reviewer #3, section 2.6.4** of the revised manuscript).
2. The analysis in AFMBench was intentionally restricted to standardized tasks as the aim was to evaluate whether LLM agents can perform routine tasks. However, we did extend the framework toward open-ended and real-world experiments (see **section 2.6** of the main manuscript).
3. Agents and tool sets were primarily developed to automate AFM measurements, evaluate the output, and interpret the results. As clarified, AILA is capable of executing any experiment that could be performed on AFM by humans. Further, the framework is flexible and could be easily extended to other applications, tools, or agents (see responses **Point 2 to Reviewer #1 and section S3** of the Supplementary Information).
4. We agree that the performance of agents are sensitive to prompts—this is a fundamental limitation of the LLM agents and not necessarily limited to AILA. To address this, we first performed prompt sensitivity analysis (see **Table S3, section 2.6.3** of the main manuscript and **Methodology**). Based on this, we identified the appropriate level of detail required in the prompt and this was consistently used across LLMs and tasks. Note that prompt optimization was not carried out on individual tasks to arrive at desirable results. Specifically, prompts were not modified or changed after performing the experiments, maintaining the integrity of the benchmark.
5. We acknowledge that the initial framework did not include an artifact detection tool and agree that this would be a valuable addition. To address this, we have now integrated two new tools, namely, Image Segment Tool & Image Scanner Tool specifically for artifact detection. Based on these tools, we designed three real-world experiments to evaluate its performance (see our response **to Point 3 of Reviewer #2**).
6. Regarding iterative debugging of the generated code, our framework already includes an agent that performs multiple rounds of debugging and provides the final, corrected code for execution. This is facilitated by the dynamic routing in AILA that evaluates the generated code through the Code Executor tool of the AFM Handler agent, and the execution is unsuccessful, recursively routes it back to the code generation task. This could also be observed from the log files of the complete trajectory shared in the Github and also the log files shared in the Supplementary Information.

These points are further elaborated in the specific questions below.

Specific questions:

1. In the evaluation of agent performance, is each task given to each model once? Or if the exact same tasks are repeated, how consistent is the agent response each time?

Response: Thank you. We agree that evaluating the performance multiple times is important for ensuring statistical validity of the results. In the original manuscript, each tasks was given only once to AILA.

To assess the performance of each model on repeated tasks and evaluate their consistency, we have now repeated each task three times. That is each experiment from AFMBench was repeated thrice on each of the models, leading to a total 1200 experiments (100 experiments x 4 LLMs x three repetition). The average values across three trials are now reported in the updated manuscript.

To address this comment, all the results in the main manuscripts have now been updated accordingly. Specifically, **Figures 3 and 4** have been updated and the corresponding text in **sections 2.3 and 2.4** have been updated. Although discussed in detail in a later comment, note that the updated Figure 3 includes **11 new metrics** to evaluate the performance of the AILA. Further, to avoid confusion between agent and tool errors, we now report them separately in **Figure 4**.

2. The current focus seems to be on the agents completing the task in one go. If an agent cannot complete the task, it knows, with appropriate prompting, to output NEED HELP to invoke the other agent. Is there a way for the user to provide additional context in such situations, and for the user's feedback to be stored to improve the agent?

Response: Excellent point. We completely agree that enabling an agent to recognize when it cannot independently solve a task and subsequently invoke assistance from another agent using the “NEED HELP” signal is crucial. In such cases, we also acknowledge that additional context or feedback provided by the user can significantly enhance the system's performance and learning. The same question was raised by Reviewer #2 (see **Point 6**).

However, the current framework has been deliberately designed to evaluate whether the system, driven by LLM agents, can autonomously solve a given task and, if needed, trigger the invocation of other agents without any human intervention. This includes allowing multiple iterations of agent invocation until either a solution is found, or concluding the process based on a predefined iteration limit. This design enables us to rigorously test the self-sufficiency and adaptability of the agentic framework under constrained conditions, where no external feedback is assumed.

While integrating user feedback mechanisms is indeed a promising direction, it falls outside the scope of the present study. Within the AILA framework, a human-in-the-loop could be easily incorporated when the system attains maximum number of iterations, for instance. However, in this case, human feedback and supervision becomes the bottle limiting the high-throughput nature of autonomous systems. Nevertheless, we agree that this is a useful framework especially for challenging and open-ended experiments and this could be pursued as part of future work.

To address this comment, we have added the following new text to **Section 2.5** of the main manuscript.

“Note that an alternate approach to implement safety is to identify risky actions and involve a human-in-the-loop mechanism to review the actions. This could also enhance the robustness and overall performance of the framework. However, such a framework requires human supervision limiting the high-throughput nature that an autonomous system can otherwise achieve, with human response becoming the

bottleneck. Hence, this approach was not implemented in AILA, and could be explored as part of future work.”

3. In Table S2, we see detailed vs. concise prompts for the same task. The concise prompt fails but the detailed prompt is a step-by-step guide for the agents. What is the advantage in using an agentic framework here when presumably an LLM with access to tools can follow instructions and execute?

Response: This is indeed an interesting question. As mentioned in response to general comments (Point 4), the goal of the experiment on prompt sensitivity was primarily to identify the level of prompt that is required for LLM agents to effectively perform the task and to understand the sensitivity to prompts. This also helped to have a principled way to develop a prompt across varying tasks.

We agree that a detailed prompt can often guide a powerful LLM to perform a task effectively when provided with the right tools. However, the primary advantage of using an agentic framework goes beyond executing a well-defined instruction. It lies in modularizing complex tasks, enabling specialized agents to collaborate, reason independently, and make dynamic decisions about which subtasks to perform, in what sequence, and with what tools. This hypothesis is further validated in the next response (**Point 4 below**), where we observe that the performance of a single-agent multi-tool framework is poorer than a multi-agent, multi-tool framework. This has also been empirically observed in our own internal analysis on other agentic frameworks as well. Essentially, multi-agent frameworks allow collaboration, cooperation, and correction enabling an overall improved performance. This observation is also consistent with several literature which demonstrates that multi-agent frameworks perform superior to single-agent frameworks²⁻⁴.

In the context of Table S2, while the detailed prompt acts as a structured guide, the agentic framework demonstrates its utility by enabling individual agents to interpret and act on subtasks autonomously, without hardcoding the entire workflow into a single prompt. This not only enhances scalability and maintainability, but also allows the system to generalize better across variations in tasks, inputs, or failure modes.

To clarify this, we have now added the following text in the Discussions (**Section 3**) of the revised manuscript along with the three new references.

“Several outcomes merit detailed examination. The measured limitations in tool coordination across different LLMs establish quantifiable thresholds for improving inter-module communication protocols. Our results demonstrate that multi-agent architectures systematically outperform single-agent configurations, with the primary advantage extending beyond mere instruction execution to encompass task modularization, specialized agent collaboration, independent reasoning, and dynamic decision-making regarding subtask sequencing and tool selection. These findings align with established literature demonstrating the superiority of multi-agent frameworks over single-agent implementations across diverse computational domains⁴⁶⁻⁴⁸.”

4. Relatedly, the structure of the framework seems to be two agents, which each have

access to two tools. How does the performance compare to if one agent was given all tools? Or if there was an agent for each tool?

Response: Good question. To evaluate the impact of framework structure on performance, we have selected a subset of 10 representative questions (see Table 2) from AFMBench and tested the performance of a single-agent multi-tool framework. Each question was evaluated across three independent trials to account for variability and ensure statistical reliability of the performance outcomes.

Table 2: Serial numbers, AFMBench question numbers, and corresponding questions used in AILA framework testing.

Serial No	Question Number	Question
1	1	Set image size as $100 \times 100 \mu\text{m}$ in the AFM software.
2	13	Initiate the scanning process in the AFM software.
3	23	Open the file 'Sample.nid', extract the data corresponding to the Z-axis backward image, calculate the number of square grids present in the image, and save the Z-axis backward image as 'question_23.png'
4	25	Set P gain to 100, I gain to 6000, and D gain to 10 in the AFM software, and then capture an image.
5	33	Open the file 'Sample.nid,' extract the Z-axis forward image data, and plot it. Save the resulting plot as 'question_33.png.'
6	35	Optimize the values of the P, I, and D gains using a genetic algorithm. Do not correct the baseline.
7	48	Open the file named 'Glass_NAP.nid' from the current directory, extract the z-axis forward image data, plot it, and save the resulting plot as 'Glass_NAP.png'.
8	57	Capture an AFM image of size $100 \mu\text{m} \times 100 \mu\text{m}$ and calculate the average surface roughness.
9	59	Capture an AFM image of size $100 \mu\text{m} \times 100 \mu\text{m}$ with the filename '59_GPT-4o'. Open the latest file from the current folder and calculate the average and root mean squared value of surface roughness.
10	100	Set image size as $150 \times 150 \mu\text{m}$ and change the cantilever tip to Dyn190Al in the AFM software.1

The figure below illustrates that the performance of GPT-3.5, LLaMA-3.3, and Claude-3.5 is relatively similar, in both the single-agent multi-tool and multi-agent multi-tool frameworks. However, their accuracy remains suboptimal across these scenarios. In contrast, GPT-4o demonstrates significantly higher performance, particularly in the multi-agent multi-tool farmwork. This improvement is largely due to GPT-4o’s ability to effectively handle scenarios involving multiple tools with similar type of names and descriptions, which often cause other models to become confused and select the wrong tool. The use of a multi-agent workflow in GPT-4o allows for tasks to be split and assigned to different agents, reducing the likelihood of tool confusion provided that no two agents have overlapping or similar tools.

Figure S5: Performance comparison of different LLM models in single-agent multitool and multi-agent multitool frameworks.

To clarify this, we have now added the following text in **Section 2.3** of the revised manuscript along with the three new references.

“Single-agent vs. multi-agent AILA. To assess whether direct tool integration with AILA yields equivalent performance to the multi-agent framework, we conducted a comparative analysis recognizing that single-agent architectures offer computational advantages through reduced intermediate processing and lower latency. A representative subset of 10 questions from the AFMBench dataset was systematically evaluated across both single-agent and multi-agent multi-tool frameworks, with each question assessed through three independent trials to ensure statistical reliability and account for inherent variability. The comparative analysis revealed framework-dependent performance variations: GPT-4o demonstrated superior performance in the multi-agent configuration (70% success rate) compared to direct tool integration (58% success rate) (see Section S6 of the supplementary material for detailed results). For alternative models, performance differences were minimal, as most architectures exhibited fundamental limitations in cross-domain tasks that inherently require multi-agent coordination, regardless of framework structure. These findings indicate that while computational efficiency favours single-agent implementations, the enhanced coordination capabilities of multi-agent frameworks provide measurable performance gains for advanced models capable of complex reasoning.”

5. More generally, the distinction between tool and agent is not clear. The authors themselves seem to use the two interchangeably, see for e.g. line 701. Figure 2 and the discussion around it centers on this distinction.

Response: Thank you for pointing this out. We have revised the manuscript to clearly distinguish between agents (LLM instances responsible for reasoning and decision-making) and tools (external functions or code modules that the agents can invoke). In our work, there are only two agents, namely AFM Handler and Data Handler agents, and six tools, namely, Document Retrieval, and Code Executor (associated with AFM Handler), and Image Optimization, Image Analyzer, and Image Segment and Image Scanner (associated with Data Handler) tools.

To address this comment, the terminology has been checked and corrected throughout the text, including in line 701 and the manuscript and Figures have been updated to maintain consistency and clarity.

6. Regarding model-specific limitations, could iterative debugging improve results for cases where the failure mode is code generation? I suggest some quantification of the distribution of types of bugs and a clarification of how that may be addressed beyond needing an entirely different LLM.

Response: Thank you. We would like to clarify that the iterative debugging has already been included in the AILA framework. In the AILA framework, if a failure occurs during code generation, the system is designed to handle this through iterative debugging. Specifically, when an error is detected, AILA captures the error log and attempts to resolve the issue iteratively. To this end, the maximum number of iterations is restricted to 20 for computational efficiency.

In practice, we have observed the following two primary scenarios: (1) Iteration Limit Exhaustion, where the system terminates after 20 unsuccessful correction attempts, with persistent errors classified as code generation failures; and (2) Instruction Overreach, where AILA generates functional code that exceeds the specified requirements, demonstrating functionality beyond the original instructions—a phenomenon indicating instruction drift or algorithmic overfitting, categorized as instruction adherence errors. These errors could potentially be addressed through several approaches including: (i) domain-specific constraint validation, (ii) improved retrieval augmented generation (RAG) architectures, and (iii) incorporating formal verification protocols that provide feedback to the generative models.

The following changes have been incorporated into **Section 2.4** of the manuscript to address the comment.

“The AILA framework incorporates iterative debugging protocols to address code generation failures through systematic error resolution. Upon error detection, AILA captures comprehensive error logs and initiates iterative correction cycles, with a maximum threshold of 20 iterations established to optimize the balance between thoroughness and computational efficiency. Analysis of debugging outcomes reveals three distinct failure modes: (1) Iteration Limit Exhaustion, where the system terminates after 20 unsuccessful correction attempts, with persistent errors classified as code generation failures; and (2) Instruction Overreach, where AILA generates functional code that exceeds the specified requirements, demonstrating functionality beyond the original instructions—a phenomenon indicating instruction drift or algorithmic overfitting, categorized as instruction adherence errors. This binary classification system enables systematic characterization of failure modes while the iteration threshold ensures computational tractability without compromising debugging efficacy in autonomous laboratory operations.”

Further, following new text has been added to the Discussion (Section 3) of the manuscript.

“Additionally, despite providing direct access to comprehensive documentation and code snippets, persistent code generation errors indicate fundamental limitations in current retrieval-augmented generation frameworks. These findings necessitate the development of enhanced code generation architectures that incorporate domain-

specific constraint validation and formal verification protocols to minimize coding errors—representing an immediate opportunity for systematic improvement in autonomous laboratory reliability.”

7. There is a lot of emphasis placed on the model evaluation. At the rate of current progress, models become obsolete in a few months. E.g. Claude 3.5 and GPT-3.5 have been succeeded by newer models.

a. It would be nice to see a focus on evaluation metrics rather than specific models

Response: Thank you for the good suggestion. We agree that model-specific evaluations can become quickly outdated given the rapid pace of development. To address this, we have focused on building robust evaluation metrics that are model-agnostic and can be consistently applied across different LLMs. Specifically, to address the Reviewer’s comment, we have now designed 11 additional metrics defined as follows.

Based on the scoring, Success rate is computed as the average score (out of 100) across three iterations for all the tasks. For all the successful runs, following metrics are employed.

1. AFM Handler Calls: The average number of times the AFM Handler agent was called to resolve a task (calls per task). An elevated score may signify a greater dependence on the AFM Handler for coordination.
2. Data Handler Calls: The average number of times the Data Handler agent was called during task solving.
3. Number of Steps: The average number of discrete actions (represented by tool or agent call) taken by the AILA to arrive at a solution.
4. Total Tokens: Total number of LLM model tokens are processed on an average for each task, considering both input (prompt) and output (complete) tokens.
5. Prompt Tokens: Number of tokens utilized in the conversation’s input or instruction segment for each task.
6. Completion Tokens: The average number of tokens that the system’s answers create for all assignments.
7. Tokens per stage: This tells how many tokens were utilized on average for each stage in the process of solving a task.
8. Latency: The average amount of time (in seconds) it takes to complete a task from the first to the final step.
9. Time per Step: The average amount of time (in seconds) spent on each step, representing the pace with which the system can perform different tasks.
10. Latency per 1000 Tokens: The average amount of time (in seconds) taken to process 1000 tokens.

Further, for each model, we run three independent trials per question. This allows us to account for variability in model responses and better understand both average performance and consistency. The results are shown below.

Metrics	GPT-4o	GPT-3.5	Llama	Claude
Success Rate (%)	65 (1.19)	32.8 (1.28)	37.3 (1.29)	45.7 (1.41)
AFM Handler Calls	2.50 (1.29)	4.90 (1.93)	4.82 (1.70)	3.06 (0.21)
Data Handler Calls	0.49 (1.05)	0.00 (0.00)	0.41(1.45)	0.00 (0.00)

Number of Steps	5.96 (1.25)	9.81 (3.06)	10.21 (2.21)	6.12 (0.42)
Total Tokens	2904 (1155)	5117 (2489)	6943 (2379)	4365 (504)
Prompt Tokens	2717 (1062)	4848(2416)	6697(2320)	4002 (482)
Completion Tokens	187 (104)	268 (89)	246 (81)	363 (42)
Tokens per Step	482 (114)	499 (85)	673 (123)	712 (38)
Latency (s)	10.64 (6.27)	11.20 (5.14)	6.89 (2.35)	17.31 (4.63)
Time/Step (s)	1.81(1.28)	1.13 (0.26)	0.66 (0.18)	2.84 (0.80)
Latency/1000 Tokens (s)	3.89(2.83)	2.33 (0.59)	1.03 (0.38)	4.02 (1.24)

Table S5. Performance of AILA with different LLMs on AFMBench. Values in the parenthesis represent the corresponding standard deviation.

To address this comment, **Figure 3b** has been revised to include **11 new metrics**, and **Table S5** has been added to the Supplementary Information. All reported values now reflect the average of three trials. Further, following new text has been added in the **Methodology section** revised manuscript.

Additional evaluation metrics are defined as following. Based on the scoring, Success rate is computed as the average score (out of 100) across three iterations for all the tasks. For all the successful runs, following metrics are employed.

- 1. AFM Handler Calls: The average number of times the AFM Handler agent was called to resolve a task (calls per task). An elevated score may signify a greater dependence on the AFM Handler for coordination.*
- 2. Data Handler Calls: The average number of times the Data Handler agent was called during task solving.*
- 3. Number of Steps: The average number of discrete actions (represented by tool or agent call) taken by the AILA to arrive at a solution.*
- 4. Total Tokens: Total number of LLM model tokens are processed on an average for each task, considering both input (prompt) and output (complete) tokens.*
- 5. Prompt Tokens: Number of tokens utilized in the conversation’s input or instruction segment for each task.*
- 6. Completion Tokens: The average number of tokens that the system’s answers create for all assignments.*
- 7. Tokens per stage: This tells how many tokens were utilized on average for each stage in the process of solving a task.*
- 8. Latency: The average amount of time (in seconds) it takes to complete a task from the first to the final step.*
- 9. Time per Step: The average amount of time (in seconds) spent on each step, representing the pace with which the system can perform different tasks.*
- 10. Latency per 1000 Tokens: The average amount of time (in seconds) taken to process 1000 tokens.*

Additionally, following new text has been added to the **Section 2.3** of the revised manuscript.

“For evaluation of our multi-agent framework, all successful trials were assessed across operational, token efficiency, and performance metrics. Operational analysis revealed significant disparities in agent coordination capabilities: Llama-3.3-70B exhibited substantial tool-agent confusion, requiring an average of 10.21 steps per task, whereas GPT-4o demonstrated superior contextual grounding and agent selection efficiency with only 5.96 average steps per task. Token utilization patterns correlated directly with these operational inefficiencies, where Llama-3.3-70B

consumed the highest average prompt tokens, indicating verbose or redundant intermediate reasoning processes, while GPT-4o achieved task objectives with minimal token usage, suggesting focused and deliberate reasoning pathways. Critical deficiencies in agent disambiguation and task-instruction alignment were observed in GPT-3.5 and Claude-3.5, which failed all three trials involving the Data Handler agent. For AFM Handler operations, GPT-4o demonstrated optimal efficiency with approximately 2.5 agent calls per task, contrasting with Claude-3.5, which generated the highest completion token counts and tokens-per-step ratios, indicating excessively elaborate intermediate outputs. Performance metrics revealed substantial variation in task completion success rates: GPT-4o achieved 65% success while GPT-3.5 performed inconsistently at 32.8%. Latency analysis showed Claude-3.5 suffered the highest mean response time (17.31 seconds), whereas Llama-3.3-70B demonstrated the lowest latency (6.89 seconds). These comprehensive metrics indicate that while Llama-3.3-70B offers reduced latency, GPT-4o provides the optimal balance between operational efficiency and execution precision, establishing it as the most suitable architecture for complex multi-agent coordination in autonomous laboratory environments.”

8. One of the critical results presented is the demonstration of experimental workflows beyond single scan acquisition. In figure 5, the prompt given is "Optimize the values of the P, I, and D gains using a genetic algorithm. Do not correct the baseline." This is quite specific. Is there any analysis on whether the agentic framework could complete the task without being told to optimize the PID, but rather more generically with instructions to optimize for image clarity and/or stability between scans? Or without specifying the algorithm to be used?

Response: To address this insightful question, we have conducted an analysis using two different prompts that did not mention PID explicitly. Instead, the prompts instructed AILA to optimize more generically for image clarity and scan stability. In both cases, as shown in Figure S3, AILA successfully selected the appropriate agent and tool to perform the task.

We have added the following text in the revised version of the manuscript:
“To further verify this, we tested two different prompts with samples of distinct morphologies (see Figure S3 in supplementary materials). In both cases, AILA correctly selected the appropriate baseline correction.”

Figure S3. Demonstration of AILA's baseline correction decisions for distinct morphologies. For the first case, where the sample had small-sized features (as mentioned in the prompt by user), AILA chose to apply baseline correction, recognizing that such features typically require correction to improve signal clarity (baseline: True). In the second case, where baseline correction was not essential due to the sample morphology, AILA skipped the correction step (baseline: False).

9. The first three discussion paragraphs summarize and repeat many points that were scattered throughout the results section. I suggest a more in-depth discussion of the last two points instead, as they seem more relevant to addressing strengths (modularity) and limitations (limited functionality) of the framework.

Response: We agree. To address this comment, the Discussion and Outlook sections have now been combined into a single section and the redundant sections and sentences have now been removed. We have also expanded our analysis of the last two points, with a particular focus on the framework's modularity as a key strength and its current limitations in functionality.

The updated Discussion section of the revised manuscript is as follows.

“AILA's modular design establishes quantifiable metrics in experimental automation through systematic benchmarking. The framework's comprehensive performance metrics in AFM operations establish standards for autonomous laboratory evaluation, while AFMBench introduces reproducible protocols for systematic assessment across experimental domains. Successful execution of tasks—from automated image optimization to nanomechanical measurements—validates the framework's capabilities for sophisticated materials characterization.

Several outcomes merit detailed examination. The measured limitations in tool coordination across different LLMs establish quantifiable thresholds for improving inter-module communication protocols. Our results demonstrate that multi-agent architectures systematically outperform single-agent configurations, with the primary advantage extending beyond mere instruction execution to encompass task modularization, specialized agent collaboration, independent reasoning, and dynamic decision-making regarding subtask sequencing and tool selection. These findings align with established literature demonstrating the superiority of multi-agent

frameworks over single-agent implementations across diverse computational domains^{46–48}. These observations also establishes an empirical baseline for balancing specialized and integrated operations—a metric applicable to automation across analytical platforms, from mass spectrometers to X-ray diffractometers.

However, the observed tendency of LLM agents to exceed operational boundaries through “sleepwalking” behavior during experimental execution presents critical safety concerns for autonomous laboratory deployment. This phenomenon, documented here for the first time in autonomous experimental systems, highlights urgent development priorities in instruction alignment and operational safety protocols. Additionally, despite providing direct access to comprehensive documentation and code snippets, persistent code generation errors indicate fundamental limitations in current retrieval-augmented generation frameworks. These findings necessitate the development of enhanced code generation architectures that incorporate domain-specific constraint validation and formal verification protocols to minimize coding errors—representing an immediate opportunity for systematic improvement in autonomous laboratory reliability.

These findings suggest specific architectural improvements for next-generation autonomous laboratories. Enhanced integration protocols between specialized agents could address the observed limitations in multi-tool coordination. Similarly, dedicated code generation modules might mitigate the predominant error mode, potentially incorporating specialized scientific programming frameworks.

The implications of this work extend beyond materials characterization. The unexpected underperformance of Claude-3.5-sonnet-20241022 compared to GPT-4o highlights a critical insight: question-answer proficiency in a specific domain does not necessarily predict effectiveness in agentic implementations. Rather tool coordination capabilities of LLMs prove to be an important aspect for effective agentic implementation. Furthermore, the observed prompt fragility emphasizes the necessity for developing rigorous evaluation frameworks prior to deployment in research environments. Specifically, developing systematic and principled approaches to develop prompts, and make systems that are robust to minor variations in prompts play a crucial role in the wider acceptance of agentic systems. To this end, quantitative benchmarks such as AFMBench provide concrete guidance for implementing LLM-driven systems in experimental research settings where precision and reliability are paramount.

Applications span pharmaceutical screening, environmental monitoring, and process optimization. For instance, documented success in parameter optimization could translate directly to automated high-throughput drug screening or catalyst discovery platforms. While current limitations in code generation and tool coordination define immediate development targets, these metrics provide clear objectives for advancing autonomous scientific platforms. The path forward requires focused development in three key areas: enhanced cross-domain reasoning capabilities, robust code generation protocols, and sophisticated multi-agent coordination mechanisms. Success in these domains would enable truly autonomous scientific platforms capable of accelerating discovery across the scientific landscape.”

10. AFM parameter optimization:

a. Not clear if the genetic algorithm code was provided or written by the LLM. If the former, why not evaluate the models on their ability to write code to solve problems?

Response: AILA employs two complementary approaches when it comes to algorithm selection, each aimed at balancing robustness and adaptability. In certain scenarios, particularly for well-established calculations, AILA uses pre-defined algorithms, adjusting only the input parameters dynamically. This ensures consistency, reproducibility, and reliability across common queries. Conversely, for more exploratory or customized tasks, such as data visualization or analysis involving unique data structures, AILA dynamically generates the necessary code. This adaptive approach allows AILA to tailor the solution more precisely to user needs, though we recognize it may introduce some variability depending on task complexity. In the case of genetic algorithm, the code was provided to AILA.

It should be noted that for executing all the tasks in AFMBench, specialized code needs to be written based on the documentation. Thus, code evaluation and successful execution is part of AFMBench. That said, evaluating the abilities of AILA to write any general purpose code, such as genetic algorithm or other optimization algorithms, is beyond the scope of the present work and can be carried out as part of future.

To clarify this point, we have added the following paragraph in the **Methodology section (AILA subsection)** of the manuscript.

“AILA uses two different approaches for selecting algorithms, depending on the task. When performing standard calculations like friction or roughness analysis, AILA relies on established algorithms with adjustable input parameters. This method produces consistent and reproducible results. For exploratory work and data visualization, AILA takes a different approach by creating the code on-the-fly. This method adapts better to varying data formats and specific user requirements; however, this may introduce variability in the results depending on the complexity of the task.”

b. What happens if you add a code writer in addition to a code executor?

Response: The AILA framework includes a code writer component designed to dynamically generate code—primarily for post-processing tasks such as analysing or visualizing AFM-generated images. However, for tasks involving direct control of AFM instrumentation, such as initiating scans or adjusting hardware parameters, AILA does not generate code autonomously. Instead, it relies strictly on predefined and validated code blocks sourced from the official documentation, which are re-organized and modified as required for the specific task at hand. This design choice is intentional, dynamically generating control code poses significant safety risks, as incorrect or malformed instructions could damage sensitive hardware. To ensure operational safety and reliability, only authorized code from the instrument’s manual is encoded and reused within AILA.

To clarify this, we have added additional text in **Section 2.5** of the main manuscript, clarifying general operations and critical operations as follows.

“...we classified all the operations that could be performed on an AFM as per the instrument’s documentation into two categories. (i) General operations—these include

setting imaging parameters, controlling the tip, selecting the scanning area, and other standard tasks. (ii) Critical operations—these involve sensitive adjustments such as factory calibrations, laser alignment, piezo calibration, and thermal calibration. Even a minor coding error in critical operations could seriously damage the instrument.

We restricted the access to the documentation of critical operations in AILA, while allowing complete access to the documentation of general operations. The critical functions are limited to trained human experts. Note that general operations were selected in such a fashion that any experiment that could potentially be performed on an AFM could be carried out using some combination of these operations. Thus, the predefined documentation in AILA does not restrict any new experiment to be performed on the AFM instrument. Note that an alternate approach to implement safety is to identify risky actions and involve a human-in-the-loop mechanism to review the actions. This could also enhance the robustness and overall performance of the framework. However, such a framework requires human supervision limiting the high-throughput nature that an autonomous system can otherwise achieve, with human response becoming the bottleneck. Hence, this approach was not implemented in AILA and could be explored as part of future work.”

11. Does the current workflow call predefined code blocks from the documentation?

Response: Yes, the current AILA workflow includes the ability to call predefined code blocks derived from documentation. These blocks are for specific AFM-related tasks. An example of such a predefined code block is provided below and also included in **Section S2.2** of the revised Supplementary information.

```
""initiate/start or terminate/stop image scanning process""
import nanosurf
import time

# load application
spm = nanosurf.SPM() # or .C3000() or .CX(), or .CoreAFM()
application = spm.application
scan = application.Scan

# Start scan
# scan.StartFrameDown() # Scanning from up to down or top to bottom
scan.StartFrameUp()    # Scanning from down to up or bottom to top

# scan.stop() # Stop scan
scanning = scan.IsScanning # Check if scanning
while scanning:
    print("Scanning in progress... Remaining lines:", (scan.Lines - scan.Currentline))
    time.sleep(3)
    scanning = scan.IsScanning
print("Scanning finished")
del spm

metadata = {'Instruction': 'AFM Code to initiate/terminate image scanning'}
```

12. How can others evaluate their models or agents on this benchmark? By definition, the benchmark should be accessible to anyone.

Response: All materials needed to evaluate other models or agents on this benchmark are openly available in the GitHub repository (<https://github.com/M3RG-IITD/AILA>). This includes the dataset, evaluation scripts, detailed instructions, and example configurations to ensure accessibility and reproducibility. In the revised version (see Data and Code Availability section), we have reorganized the repository structure to make it clearer and easier to navigate. We encourage others to explore the benchmark and use it for their own evaluations.

Reviewer #5 (Remarks to the Author):

Response: Thank you for co-reviewing our manuscript and for your valuable input. We greatly appreciate your time, effort, and collaboration in helping us improve our work.

References

1. pymoo - NSGA-II: Non-dominated Sorting Genetic Algorithm.
<https://pymoo.org/algorithms/moo/nsga2.html>.
2. Shu, R., Das, N., Yuan, M., Sunkara, M. & Zhang, Y. Towards Effective GenAI Multi-Agent Collaboration: Design and Evaluation for Enterprise Applications. Preprint at <https://doi.org/10.48550/arXiv.2412.05449> (2024).
3. Gao, M. *et al.* Single-agent or Multi-agent Systems? Why Not Both? Preprint at <https://doi.org/10.48550/arXiv.2505.18286> (2025).
4. Sreedhar, K. & Chilton, L. Simulating Human Strategic Behavior: Comparing Single and Multi-agent LLMs. Preprint at <https://doi.org/10.48550/arXiv.2402.08189> (2024).

Manuscript Ref. ID: NCOMMS-25-14397A

Title: Autonomous Microscopy Experiments through Large Language Model Agents

We have carefully addressed all the minor comments provided by the reviewers, and have incorporated the corresponding revisions into the main manuscript, with changes highlighted in blue. Below, we present a point-by-point response to each comment.

REVIEWER COMMENTS

Reviewer #2 (Remarks to the Author):

I believe the quality of the manuscript has improved substantially after the initial review.

Response: We appreciate your positive feedback.

Q1: On lines 279-280, the manuscript mentions that single-agent architectures offer computational advantages through reduced intermediate processing and lower latency. However, there is no specific data or evidence provided to support this claim.

Response: We thank the reviewer for the careful reading and observation. To address this comment, the statement has now been removed from the manuscript.

Q2: On line 271, the manuscript introduces the concepts of single-agent and multi-agent systems. However, in the subsequent sections, particularly lines 277-292, there seems to be some confusion regarding whether the same concepts are being discussed. Additionally, the data in Figure 3d and Figure S5 appear to be inconsistent, which suggests that these two sections might be referring to different concepts under the same terminology. Could the authors please clarify the relationship between these concepts and ensure consistency in the terminology used throughout the manuscript?

Response: Thank you for highlighting the concern regarding potential inconsistency in the terminology. We have defined two distinct AILA frameworks:

1. **Single-agent architecture**, where all tools are connected to a single agent.
2. **Multi-agent architecture**, where tools are divided among multiple agents, each handling a subset of the tasks.

We evaluated both architectures using a representative subset of 10 questions from the AFMBench dataset. The results are presented in Figure S5 and the corresponding questions are listed in Table S3 of the supplementary material. Based on this evaluation, the multi-agent architecture demonstrated superior performance.

Given this outcome, we further tested the multi-agent architecture on the full AFMBench dataset. To better interpret the results, we categorized the tasks based on three criteria:

1. Complexity level (Basic vs. Advanced)
2. Number of tools required (Single vs. Multiple)
3. Type of agent configuration needed (Single-agent vs. Multi-agent)

The performance breakdown across these categories is presented in Figure 3d.

To eliminate confusion, we have now adopted consistent terminology throughout the revised manuscript—using “**single-agent architecture**” and “**single-agent task**” (as well as their multi-agent counterparts) to clearly differentiate between system design and task characteristics.

To address this comment, we have now revised the manuscript to clarify and consistently distinguish between these concepts. Additionally, the following figures have been updated in the revised version of the manuscript.

Figure 3. Performance comparison of different models across tasks of varying complexity (Advanced/Basic) and requiring different tools (Single/ Multiple) and agents (Single/Multiple).

Figure S5: Performance comparison of different LLM models in single-agent and multi-agent architecture.